# Enhancer-associated long non-coding RNA LEENE regulates endothelial nitric oxide synthase and endothelial function

Yifei Miao[1], Nassim E. Ajami[2], Tse-Shun Huang[3], Feng-Mao Lin[1], Chih-Hong Lou[4], Yun-Ting Wang[1], Shuai Li[5], Jian Kang[5], Hannah Munkacsi[4], Mano R. Maurya[3], Shakti Gupta[3], Shu Chien[3,5], Shankar Subramaniam[3,6] & Zhen Chen [1]

The optimal expression of endothelial nitric oxide synthase (eNOS), the hallmark of endothelial homeostasis, is vital to vascular function. Dynamically regulated by various stimuli, eNOS expression is modulated at transcriptional, post-transcriptional, and post-translational levels. However, epigenetic modulations of eNOS, particularly through long non-coding RNAs (lncRNAs) and chromatin remodeling, remain to be explored. Here we identify an enhancer-associated lncRNA that enhances eNOS expression (LEENE). Combining RNA-sequencing and chromatin conformation capture methods, we demonstrate that LEENE is co-regulated with eNOS and that its enhancer resides in proximity to *eNOS* promoter in endothelial cells (ECs). Gain- and Loss-of-function of LEENE differentially regulate eNOS expression and EC function. Mechanistically, LEENE facilitates the recruitment of RNA Pol II to the *eNOS* promoter to enhance eNOS nascent RNA transcription. Our findings unravel a new layer in eNOS regulation and provide novel insights into cardiovascular regulation involving endothelial function.

[1] Department of Diabetes Complications and Metabolism, Beckman Research Institute, City of Hope, 1500 Duarte Rd., Duarte, CA 91010, USA. [2] Bioinformatics and Systems Biology Graduate Program, University of California at San Diego, 9500 Gilman Dr., La Jolla, CA 92093, USA. [3] Department of Bioengineering and Institute of Engineering in Medicine, University of California at San Diego, 9500 Gilman Dr., La Jolla, CA 92093, USA. [4] Genome Editing Core, Beckman Research Institute, City of Hope, 1500 Duarte Rd., Duarte, CA 91010, USA. [5] Department of Medicine, University of California at San Diego, 9500 Gilman Dr., La Jolla, CA 92093, USA. [6] Departments of Cellular and Molecular Medicine and Computer Science and Engineering, University of California San Diego, 9500 Gilman Dr., La Jolla, CA 92093, USA. Yifei Miao, Nassim E. Ajami, and Tse-Shun Huang contributed equally to this work. Correspondence and requests for materials should be addressed to Z.C. (email: zhenchen@coh.org) or to S.S. (email: shankar@ucsd.edu) or to S.C. (email: shuchien@ucsd.edu)

Endothelial nitric oxide synthase (eNOS), which is central to endothelial homeostasis and vascular function, is regulated at multiple levels[1], including post-translational modifications (such as phosphorylation and acetylation)[2, 3] and transcriptional regulation by transcription factors (TFs)[4]. It has been established that eNOS transcription is largely regulated by Krüppel-like factors 2 (KLF2) and 4 (KLF4), two key TFs in endothelial identity and vascular homeostasis[5]. The expression and activity of KLF2 and KLF4 can be altered by a number of mechanical (e.g., hemodynamic flow), biochemical (e.g., pro-inflammatory stress), and pharmacological stimuli (e.g., cardiovascular protective drugs), leading to differential transcriptional regulation of eNOS as well as other genes important in endothelial biology[6, 7]. There is also evidence that eNOS expression can be regulated through histone modifications[8, 9]. However, whether and how long-range DNA interaction coordinates with TF binding and histone modification to modulate eNOS transcription in endothelial cells (ECs) remains essentially unknown.

Long-non-coding RNAs (lncRNAs) are a large class of ncRNAs that are >200 bp in length. Over 27,000 lncRNAs have been predicted/annotated in the human genome[10], but relatively little is known about their biological functions and the classification can be ambiguous due to the lack of functional characterization[11]. Depending on their subcellular localization (i.e., in the nucleus or cytoplasm), lncRNAs can regulate gene expression through diverse mechanisms. A group of lncRNAs has been identified as nucleus-retained and chromatin-associated[12–15]; they can serve as scaffolds or guides in cis or in trans to recruit TFs, transcriptional co-activators, or chromatin remodelers, and/or to promote long-range DNA (e.g., promoter-enhancer) interaction, thus resulting in transcriptional activation[16–18]. For example, lncRNA Firre has been shown to be localized around its site of transcription in X-chromosome in the embryonic stem cells and mediate trans-chromosomal interaction[18]. LncRNAs can also be classified depending on their encoded genomic locations (i.e., intragenic, intergenic, or enhancer regions) and the associated histone modifications[11]. A new class of lncRNAs have emerged as lnc-eRNA or elncRNA, which are encoded in enhancer regions marked by histone 3 lysine 4 monomethylation (H3K4me1) and histone 3 lysine 27 acetylation (H3K27ac)[19, 20]. The regulatory role of this new class of lncRNAs, especially those in the vascular ECs, has not been explored.

In this study, we investigate the role of lncRNAs in endothelial homeostasis using the endothelial hallmark eNOS as a prototype. In the characterization of endothelial responses to physiological and pathophysiological conditions, ECs subjected to different flow patterns offer an excellent model to investigate the epigenetic mechanisms underlying the distinct gene expression profiles and the consequent opposing functional phenotypes[21]. For example, the transcriptomes and DNA methylomes of ECs subjected to flow have begun to be characterized[22–25]. Herein, by combining transcriptome and chromatin conformation profiling, we identify a lncRNA that is transcribed from an enhancer that has proximal association with eNOS genomic locus. Furthermore, the lncRNA transcript serves as a guide to enhance eNOS transcription through the recruitment of RNA polymerase II (Pol II) and nascent messenger RNA (mRNA) transcription. We hence termed it "lncRNA that enhances eNOS expression" (LEENE). Using multiple gain- or loss-of-function approaches, we provide evidence that LEENE promotes eNOS transcription, eNOS-derived NO bioavailability, and endothelial function.

## Results

**LEENE is highly co-regulated with eNOS in ECs.** To identify the lncRNAs that potentially regulate eNOS, we recently profiled the transcriptome using poly A-selected RNA-seq in ECs subjected to physiological flow with pulsatile shear stress (PS) and pathological flow with oscillatory shear stress (OS) for 10 different time durations ranging from 1 to 24 h[26]. We analyzed the lncRNA profile following the pipeline illustrated in Fig. 1a. Among the 2054 lncRNAs identified in the RNA-seq, we first filtered for those differentially regulated by PS vs. OS at the end point (i.e., "h 24"). These flow-regulated lncRNAs are listed and ranked based on their differential expression in the heatmap in Fig. 1b. We then sought for lncRNAs positively correlated with eNOS over the entire time course (correlation coefficient >0.8). The RNA level of LEENE (gene name LINC00520, aka C14orf34), which is the top-ranked candidate, was upregulated to ~4-fold by PS as compared with OS by 24 h, and it was highly correlated with that of eNOS (correlation coefficient 0.85, Fig. 1c). The temporal course of flow-regulated eNOS and LEENE RNA levels showed similar patterns, i.e., a sustained and robust induction by PS. Their RNA levels peaked at 9 h and continued to be elevated up to 24 h. Notably, mRNAs encoding KLF2 and KLF4, the key TFs of eNOS, were significantly induced by PS as early as 1 h, reached their highest levels at 4–6 h, and remained induced at 24 h. In contrast to the PS induction of LEENE and KLF-eNOS signaling, the pro-inflammatory vascular cell adhesion molecule 1 (VCAM1) was suppressed by PS (Fig. 1d).

Exemplified by the RNA-seq data from "h 24" (Fig. 1d), LEENE has two transcripts, which are encoded by Exons 1–4 (the less abundant form in ECs) and Exons 1, 3, and 4 (the predominant form). By referring to FANTOM5[27], we confirmed that neither LEENE transcripts has any coding potential. Using absolute quantification assay, we determined the copy number of LEENE to be ~10 copies per cell in untreated HUVECs and ~40 copies per cell in PS-treated HUVECs. Under PS, both LEENE transcripts were upregulated as compared with OS (Fig. 1e). As shown in Fig. 1f, quantitative PCR (qPCR) with LEENE RNA-specific primers also revealed the significantly higher level of LEENE in ECs subjected to PS than to OS. To confirm the flow regulation of LEENE in ECs and explore its relevance to endothelial function, we tested whether LEENE is differentially regulated by tumor necrosis factor alpha (TNFα), which exerts pro-inflammatory effects similar to OS, and atorvastatin (ATV), which confers endothelial protective effects similar to PS. Resembling the opposite effects of OS and PS, TNFα decreased, while ATV increased the level of LEENE. These findings are in line with the differential levels of KLF2/KLF4-eNOS signaling (Fig. 1g).

The similar regulation of LEENE and eNOS prompted us to examine whether LEENE is also a transcriptional target of KLF2 and/or KLF4. To test this possibility, we first attempted to identify the promoter/enhancer region of LEENE. In evaluating the genomic region surrounding LEENE, i.e., 20 kb up- and downstream of the transcription start site (TSS) on chr14: 56,245,000–56,285,000, we observed the enrichments of H3K27ac and H3K4me1 in the HUVEC chromatin immunoprecipitation (ChIP)-seq data available in the ENCODE Database, indicating an "enhancer" state of this DNA region (Fig. 2a). Because KLF2/KLF4 can transactivate eNOS through TF-binding sites (TFBS) in the eNOS promoter regions[28], we next searched for TFBS in the DNA region in and near LEENE locus. As illustrated in Fig. 2a, the region spanning −20 to +5 kb of LEENE TSS contains multiple TFBS for KLF2 and KLF4. We subsequently overexpressed KLF2 and KLF4 in ECs to experimentally verify whether these key TFs can upregulate LEENE. Indeed, we found increased levels of LEENE by the overexpression of KLF2 or KLF4 in ECs, with eNOS as a positive control (Fig. 2b, c). To confirm the association of such TFs on the promoter of LEENE, we performed ChIP-qPCR, which detected a robust binding

between KLF4 and multiple regions within the promoter region of *LEENE* (marked by H3K4me3 peaks, Fig. 2a); these interactions were significantly increased by Ad-KLF4, which mimics the effect of PS and ATV (Fig. 2d). Complementarily, when we knocked down KLF2 in ECs, the PS-regulated LEENE was substantially decreased (Fig. 2e). Collectively, the data in Figs. 1 and 2 suggest that LEENE is (1) co-regulated with eNOS downstream of KLF2 and KLF4, and (2) induced in conditions that promote endothelial homeostasis but suppressed by stimuli that impair endothelial function.

**Proximal association between *LEENE* and *eNOS* genomic loci.** To gain insights into the biological function of LEENE, we first determined its subcellular localization. As shown in Fig. 3a, LEENE RNA transcripts were predominantly detected in the nucleus of ECs, i.e., as chromatin-associated and nucleoplasm-localized, with only a minor fraction in the cytoplasm, suggesting that its biological function is mainly in the nucleus. To this end,

we used MALAT1 as a positive control, which has been previously identified as nucleus-enriched lncRNA[14]. We also used DANCR as the cytoplasm-enriched lncRNA control and CasC7 and TUG1 as the controls for lncRNAs localized in both nucleus and cytoplasm[29, 30] (Supplementary Fig. 1). We then assessed the genomic features and neighboring genes of *LEENE*. As recently described, *LEENE* is located 110 kb downstream of *KTN1* and 321 kb upstream of *PELI2*[31]. Unlike LEENE, the mRNA levels of neither KTN1 nor PELI2 were differentially regulated by PS or OS in ECs (Supplementary Fig. 2a). These results suggest that LEENE is transcribed independently from its neighboring genes. The strong enhancer marks (i.e., H3K27ac and H3K4me1 peaks) surrounding LEENE including the 5′, gene body, and 3′ regions (Fig. 3b) suggest that *LEENE* genomic locus may act as a distal enhancer to mediate transcriptional activation in ECs. We did find that PS led to significant increase in the H3K27ac in the *LEENE* region as measured by ChIP-qPCR, indicating the activation of *LEENE* as an enhancer in ECs subjected to PS vs. OS (Fig. 3c).

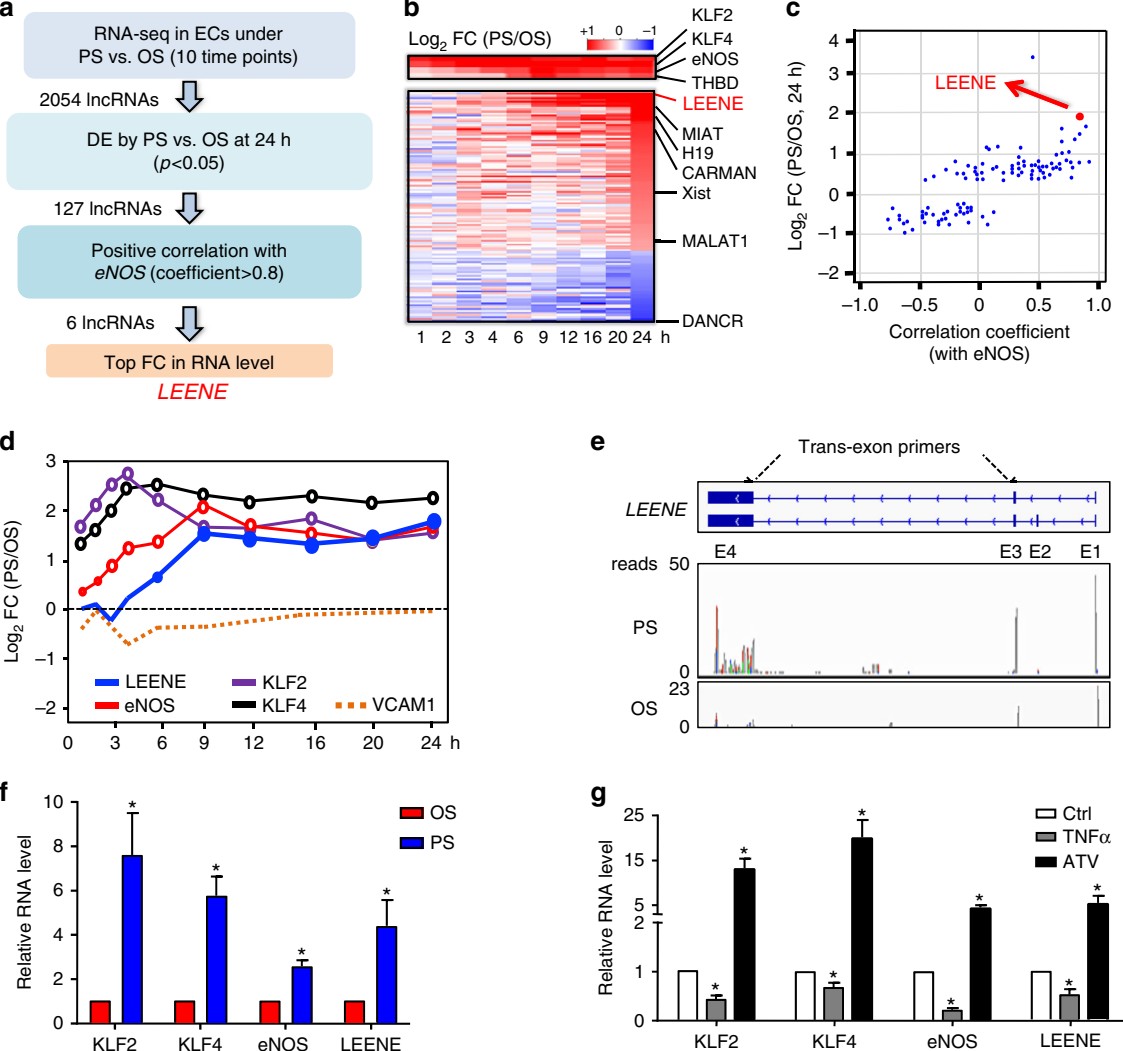

**Fig. 1** Co-regulation of LEENE and eNOS. **a** LEENE discovery pipeline. **b** Heatmap of RNA levels of flow-regulated lncRNAs derived from RNA-seq. **c** Scatter plot of the flow-regulated lncRNAs ranked by differential expression (DE) fold change (FC) at 24 h (PS/OS) and correlation with eNOS mRNA level. **d** Time course of log₂FC of mRNAs encoding various genes. **e** Structure of *LEENE* gene encoding two RNA transcripts. Trans-exon primers used in qPCR were designed to amplify fragments flanking Exons 3 and 4. RNA-seq tracks depicting abundance of LEENE in ECs under PS or OS for 24 h. **f**, **g** qPCR detection of various RNA transcripts in ECs subjected to PS or OS (in **f**) or TNFα (100 ng per ml) or atorvastatin (ATV) (1 μM) (in **g**) for 24 h. Data are presented as mean ± SEM, *n* = 5 in each group. *\*p* < 0.05 compared to OS (in **f**, based on Student's *t* test) or ctrl (in **g**, calculated by ANOVA followed by Bonferroni post test)

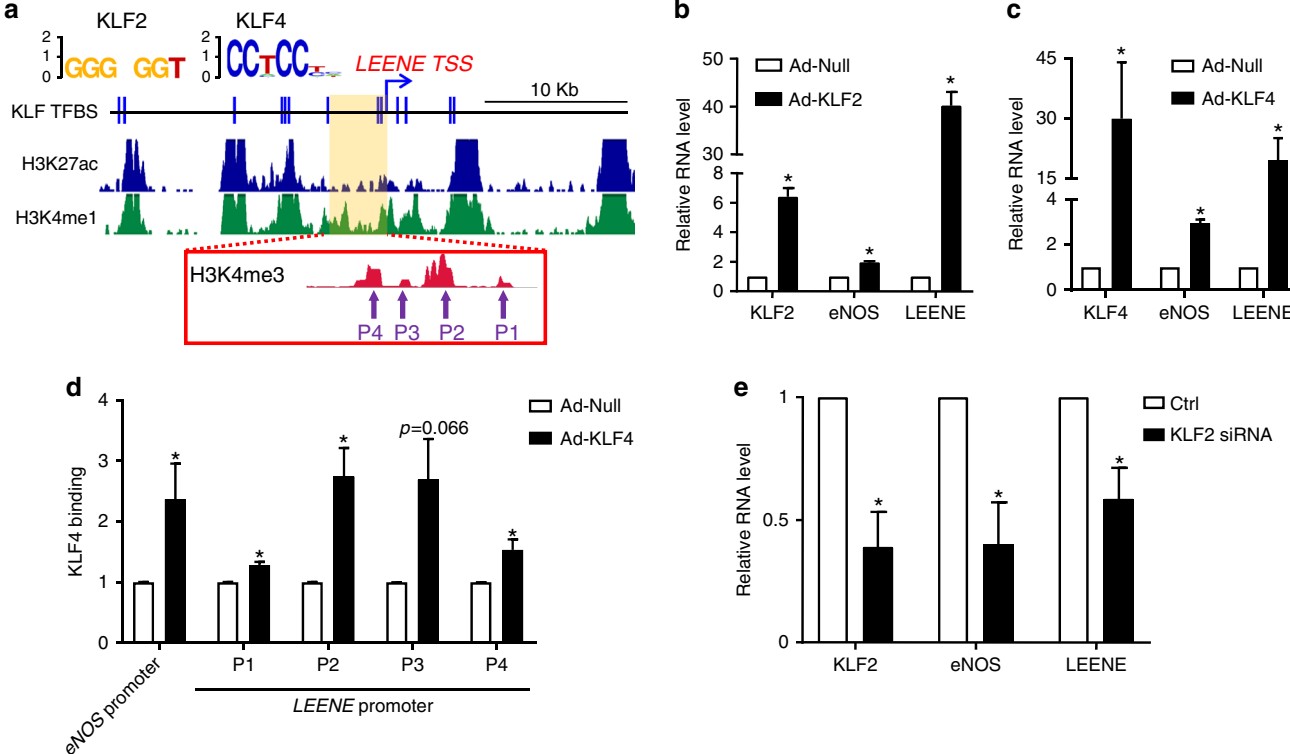

**Fig. 2** KLF2 and KLF4 transcriptionally regulate LEENE. **a** Putative KLF2 and KLF4 binding sites in *LEENE* enhancer/promoter based on the conserved KLF2 and KLF4 binding motifs (shown on the top). Middle tracks display H3K27ac and H3K4me1 ChIP-seq signals in *LEENE* locus, and the inset shows H3K4me3 ChIP-seq signals in the putative *LEENE* promoter region from ENCODE HUVEC data; arrows indicate regions detected in ChIP-qPCR (Fig. 2d). **b–d** HUVECs were infected with respective adenoviruses for 48 h. RNA levels of KLF2, KLF4, LEENE, and eNOS were detected by qPCR (in **b** and **c**) and KLF4 binding to promoters of *eNOS* and *LEENE* was quantified by ChIP-qPCR analysis (in **d**). **e** qPCR of respective RNA levels in ECs transfected with scramble control (Ctrl) or KLF2 siRNA. Data are presented as mean ± SEM, $n = 5$ in each group. *$p < 0.05$ compared to respective controls using Student's $t$ test

To explore the potential function of *LEENE* in chromatin remodeling, we took advantage of the HUVEC Hi-C data available at Gene Expression Omnibus (GSE63525)[32, 33]. First, we mapped all the potential inter-chromosomal interactions with *LEENE* locus (i.e., ~50 kb H3K27ac-enriched enhancer region illustrated in Fig. 3b) and identified 2794 genes encoded in these potential interacting regions (Fig. 3d). Among these genes, 1177 showed differential expressions in the time series RNA-seq profiles from HUVECs subjected to PS vs. OS (Fig. 3d). Upon examining the correlation of LEENE with these genes using time series RNA-seq profiles, we identified 81 genes that are highly correlated with LEENE, and eNOS was among the top hit with the highest correlation (Supplementary Fig. 3). The inter-chromosomal interaction between *eNOS* (chr7: 150,700,000–150,705,000) and *LEENE* (chr14: 56,280,000–56,285,000) in HUVEC is illustrated in Fig. 3e. Of note, such interaction is absent in human epithelial and HeLa cells (Supplementary Fig. 4), which do not express detectable level of endogenous eNOS.

To confirm the proximal association between *eNOS* and *LEENE*, we performed DNA fluorescence in situ hybridization (FISH), which has been commonly used to validate the chromosomal association revealed by chromatin conformation capture-based methods[34]. Indeed, we observed the proximity association of *eNOS* and *LEENE* probes in ~10% of ECs under 24 h PS, the physiological flow condition (Fig. 3f). To further confirm and quantitatively compare the *LEENE–eNOS* inter-chromosomal interaction in ECs under different flow conditions, we performed high-resolution 4C-seq in HUVECs subjected to PS and OS using the H3K27ac- and H3K4me1-enriched peak region

in the *eNOS* promoter as the bait (Supplementary Fig. 5). This region was previously identified to be crucial for endothelial-specific eNOS expression[8]. Consistent with the Hi-C data, 4C-seq also revealed the chromosomal proximity between the *LEENE* enhancer and *eNOS* promoter and this interaction is substantially increased in ECs subjected to PS as compared with OS (Fig. 3g).

**LEENE enhances eNOS expression through chromatin association.** To examine whether the *LEENE*-associated enhancer plays a role in positive regulation of eNOS transcription, we employed CRISPR-Cas9 gene editing to remove the ~10 kb enhancer region of *LEENE* immediately upstream of its TSS, as illustrated in Fig. 4a. The single-guide RNAs (sgRNA)-guided Cas9 cutting efficiency was first verified using the surveyor assay in human embryonic kidney (HEK) 293 cells and then in ECs using genomic PCR assay with primers probing the 5′ and 3′ ends of the targeted region (Supplementary Fig. 6). As a result of the enhancer ablation, the transcription of LEENE and eNOS was significantly suppressed, in both DMSO (a control vehicle) and ATV-treated ECs (Fig. 4b). These changes in gene expression were attendant with similar changes in the proximity association between *eNOS* and *LEENE*, as revealed by DNA FISH (Fig. 4c, d). We also deleted the coding region of *LEENE* in ECs and examined the eNOS expression in ECs. As shown in Fig. 4e, the deletion of *LEENE* coding region also significantly decreased eNOS mRNA level. Taken together, the *LEENE* enhancer forms proximity association with *eNOS* promoter to serve as a prerequisite for eNOS expression under both untreated and statin-induced conditions. It is to be noted that LEENE RNA transcript

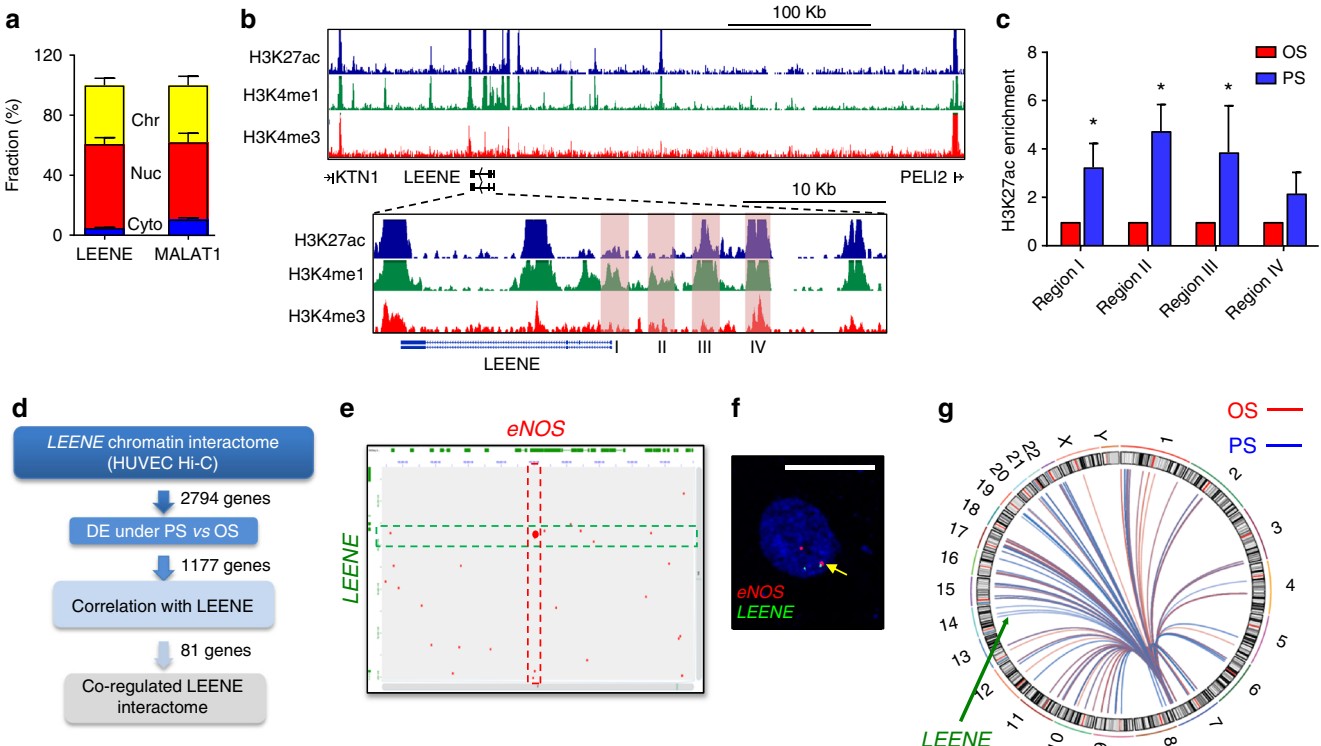

**Fig. 3** LEENE RNA is nucleus-localized and its DNA lies in enhancer region interacting with *eNOS* promoter. **a** qPCR quantitation of LEENE and MALAT1 in subcellular fractions from ECs, plotted as percentages in association with chromatin (Chr), nucleoplasm (Nuc), and cytoplasm (Cyt). **b** ENCODE HUVEC ChIP-seq signals in 400 kb (top tracks) and 50 kb (bottom tracks) regions surrounding *LEENE*. Regions in shades were selected for H3K27ac ChIP-qPCR in **c**. **d** Flow chart of integrative Hi-C and RNA-seq analyses. **e** *LEENE–eNOS* interaction map generated from GEO HUVEC Hi-C analysis. Red pixels represent interactions between two regions, respectively, in chr7 (*X*-axis) and chr14 (*Y*-axis). The highlighted regions correspond to *eNOS* promoter and *LEENE* enhancer regions. **f** Representative image of DNA FISH with respective probes recognizing *LEENE* and *eNOS* genomic loci. Arrow indicates proximity association between two loci. Scale bar = 10 μm. **g** 4C-seq mapping of inter-chromosomal interactions between *eNOS* bait (249 bp) and lncRNAs listed in Fig. 1b. Each line in the circoplot represents an interaction and the color intensity reflects the normalized reads of ligated DNA ends. Chromosomes are numbered around the circle. Data are presented as mean ± SEM, *n* = 5 in each group. \**p* < 0.05 compared to OS based on Student's *t* test

may also mediate, at least in part, this positive regulation of eNOS.

**LEENE RNA enhances eNOS transcription and EC function.** To further address the role of LEENE RNA transcript, we inhibited LEENE using LNA Gapmers, which can effectively silence the target nuclear RNA via an RNase H-mediated degradation[35]. First, we tested two LNAs targeting two different regions of LEENE in ECs under basal condition. Compared with the scrambled control, both LEENE-inhibiting LNAs decreased the basal eNOS mRNA levels in HUVECs (Fig. 5a). To confirm this result, we also silenced LEENE in human aortic ECs (HAoECs), i.e., ECs with a different origin, and observed a similar effect in the suppression of eNOS mRNA expression (Supplementary Fig. 7). In addition to the suppressive effect on eNOS, LEENE LNAs led to an increased transcription of pro-inflammatory molecules intercellular adhesion molecule 1 (ICAM1) and VCAM1 (Supplementary Fig. 8). We further demonstrated that the inhibition of LEENE RNA decreased eNOS expression at the protein level in ECs in response to pharmacological or physiological stimuli, i.e., ATV or PS (Fig. 5b; Supplementary Fig. 9). To examine the functional regulation of LEENE in ECs, we performed monocyte adhesion assay to assess the effect of LEENE blockade on eNOS-mediated anti-inflammatory function. As shown in Fig. 5c, d, inhibition of LEENE significantly increased the number of monocytes adhering to ECs subjected to PS.

To mimic the effect of LEENE induction by PS, ATV, and KLF2/KLF4, we overexpressed LEENE in its predominant form (encoded by Exons 1, 3, and 4) in ECs using a CMV-driven and GFP-tagged adenovirus. With comparable transfection efficiency as control GFP vector, LEENE overexpression increased the mRNA levels of eNOS in both HUVECs and HAoECs (Fig. 5e; Supplementary Fig. 10). In line with the increased eNOS transcription, LEENE overexpression also led to increased eNOS protein level (Fig. 5f; Supplementary Fig. 9) and eNOS-derived NO production (Fig. 5g). Collectively, the results in Fig. 5 suggest that LEENE RNA positively regulates eNOS expression and its associated endothelial function.

**LEENE RNA promotes eNOS nascent mRNA transcription.** We next examined the molecular mechanism that explains how LEENE promotes the eNOS transcription. Because enhancer-promoted transcription typically requires TFs, mediator (Med) complex[36] and RNA Pol II[37], and lncRNAs have been suggested to bind these factors/complexes to promote transcription[19,38], we hypothesized that LEENE RNA transcript may promote eNOS transcription by facilitating the recruitment of one or more of these transcriptional activators in the *LEENE–eNOS* loci.

We first tested whether there is an increased binding between LEENE and these TFs in ECs treated with ATV. As shown in Fig. 6a–c, RNA-IP revealed that the associations of LEENE RNA with Pol II, KLF4, and MED1, and were substantially enhanced in ECs treated with statin. As an isotype control, IgG did not pull

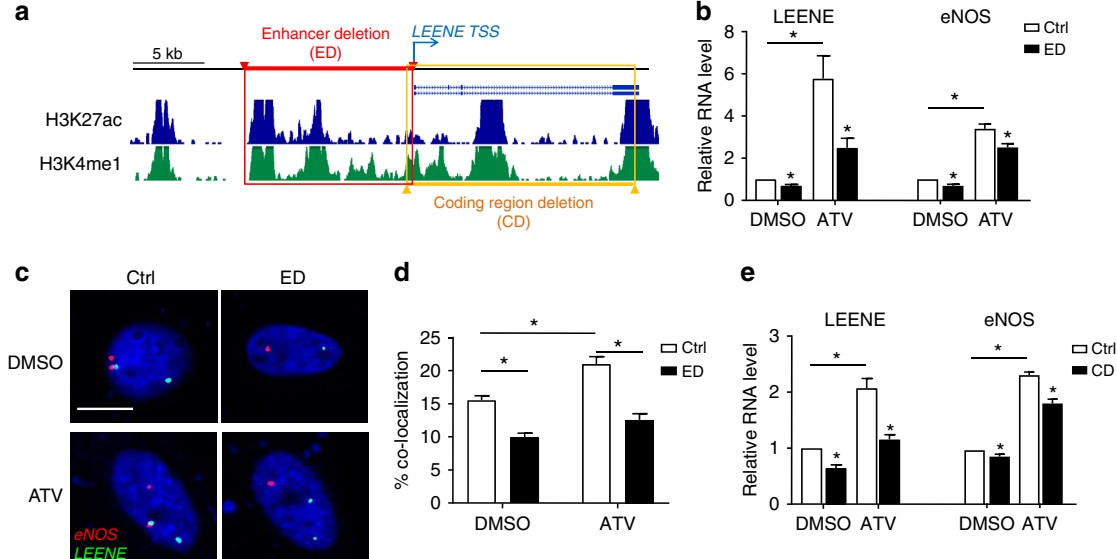

**Fig. 4** Gene editing of *LEENE* locus influences eNOS transcription. **a** Schematic illustration of CRISPR-Cas9 targeting strategy. Regions in red and orange indicate, respectively, the upstream enhancer/promoter region or the coding region deleted by sgRNA-guided Cas9, resulting in enhancer deletion (ED) and coding region deletion (CD) in *LEENE* locus. **b** LEENE and eNOS RNA levels in ECs transfected with control Cas9 plasmid (Ctrl) or "ED" Cas9-sgRNAs were quantified using qPCR. **c** DNA FISH for proximity association of *LEENE* and *eNOS* genomic loci. ECs transfected with control (Ctrl) or "ED" Cas9-sgRNAs were treated with DMSO or ATV (1 μM) for 24 h. Scale bar = 10 μm. **d** Percentage of cells with *LEENE* and *eNOS* proximity association (distance <1 μm). $n = 678$ in "DMSO-Ctrl" group; $n = 425$ in "DMSO-ED" group; $n = 632$ in "ATV-Ctrl" group; $n = 581$ in "ATV-ED" group. **e** qPCR quantification of LEENE and eNOS RNA levels in ECs transfected with control Cas9 (Ctrl) or "CD" Cas9-sgRNAs. All data are presented as mean ± SEM. $n = 5$ in each group unless specified. *$p < 0.05$ compared with "Ctrl" or between indicated groups based on Student's $t$ test

down significant amount of LEENE RNA, and the association of LEENE RNA with IgG was not altered by statin treatment (Supplementary Fig. 11). In order to test whether LEENE RNA associate with *LEENE–eNOS* loci, we performed chromatin isolation by RNA purification (ChIRP) assay with two pools of biotin-labeled RNA probes (even-numbered and odd-numbered), each with five probes containing sequences complementary to the respective regions of LEENE (Fig. 6d). We were able to recover/enrich LEENE RNA specifically and efficiently, with β-actin RNA as a negative control (Supplementary Fig. 12a). Of note, this enrichment of LEENE was not achieved with biotin-labeled LacZ probes (Supplementary Fig. 12b). In the LEENE-enriched chromatin precipitates, the DNA sequences in *LEENE* enhancer region and *eNOS* promoter were also detected, suggesting that LEENE RNA indeed interact with these chromosomal regions (Fig. 3e). Furthermore, these interactions were increased by statin treatment, which induces eNOS and LEENE (Fig. 3e). As an additional control, we performed ChIRP assay in HEK293 cells, which do not express detectable level of endogenous eNOS. Compared to ChIRP performed using ECs, LEENE ChIRP using HEK293 cells revealed virtually no binding between LEENE RNA and the genomic loci of *LEENE* and *eNOS* (Supplementary Fig. 12c). Importantly, the statin-induced interaction between LEENE RNA and *eNOS* locus appears to be region-specific because this was absent for the 150 kb up- or downstream of *LEENE* encoding *PELI2* and *KTN1*, respectively (Supplementary Fig. 12d).

Next, to test whether LEENE is required for the recruitment of KLF2/KLF4, Med1, and RNA Pol II to enhance eNOS transcription, we determined the association of these proteins with the *eNOS* promoter in LEENE-depleted cells. As shown in Fig. 6f, compared with ECs transfected with scramble LNA, LEENE LNA resulted in a reduced association between RNA Pol II and multiple *eNOS* promoter regions, while that between KLF4 or Med1 and *eNOS* promoter regions did not change (Supplementary Fig. 13). In line with the inhibitory effect of LEENE LNA on

Pol II binding to *eNOS* promoter, LEENE LNA caused a significant decrease in nascent eNOS mRNA level, which was quantified by nascent RNA pulldown combined with qPCR (Fig. 6g). Collectively, Fig. 6 suggests that LEENE RNA regulates the transcription of *eNOS* gene by facilitating the recruitment of RNA Pol II and the resultant nascent RNA transcription.

**Mouse homolog of human LEENE**. Next, we explored the conservation of LEENE between human and mouse. First, we compared the genomic structure of chromosomal region between *KTN1* and *PELI2* in human vs. mouse and found an expressed sequence tag (EST) (identifier BY707159.1) located in the similar region in mouse chromosome 14 as LEENE. Similar to the human LEENE, BY707159.1 is also transcribed from the negative strand (Fig. 7a), and the surrounding DNA region contains multiple KLF2/KLF4 binding sites (Fig. 7b). Comparison of the sequences of BY707159.1 showed that 472 out of 680 bp were aligned to the Exons 1, 3, and 4 of the human LEENE (Fig. 7c). To explore its functional and disease relevance, we examined the level of BY707159.1 in the mouse artery. It is well established that the mouse thoracic aorta (TA) and aortic arch (AA) are associated, respectively, with distinct flow patterns and opposite endothelial phenotypes, and that eNOS is expressed at a significantly higher level in TA than AA[39]. Hence, we determined the transcription level of BY707159.1 in TA and AA isolated from C57BL mice. As shown in Fig. 7d, the level of BY707159.1 was ~8 fold higher in TA than AA; this recapitulates the PS induction and OS suppression of LEENE levels in the human ECs (Fig. 1). Further, we have isolated lung ECs from C57BL mice and examined the potential regulation of eNOS by LEENE homolog in mouse. As shown in Fig. 7e, LNA inhibiting of LEENE homolog indeed significantly decreased the mRNA level of eNOS in the mouse lung ECs. These findings suggest that LEENE regulation of eNOS may be a conserved mechanism in mouse and human.

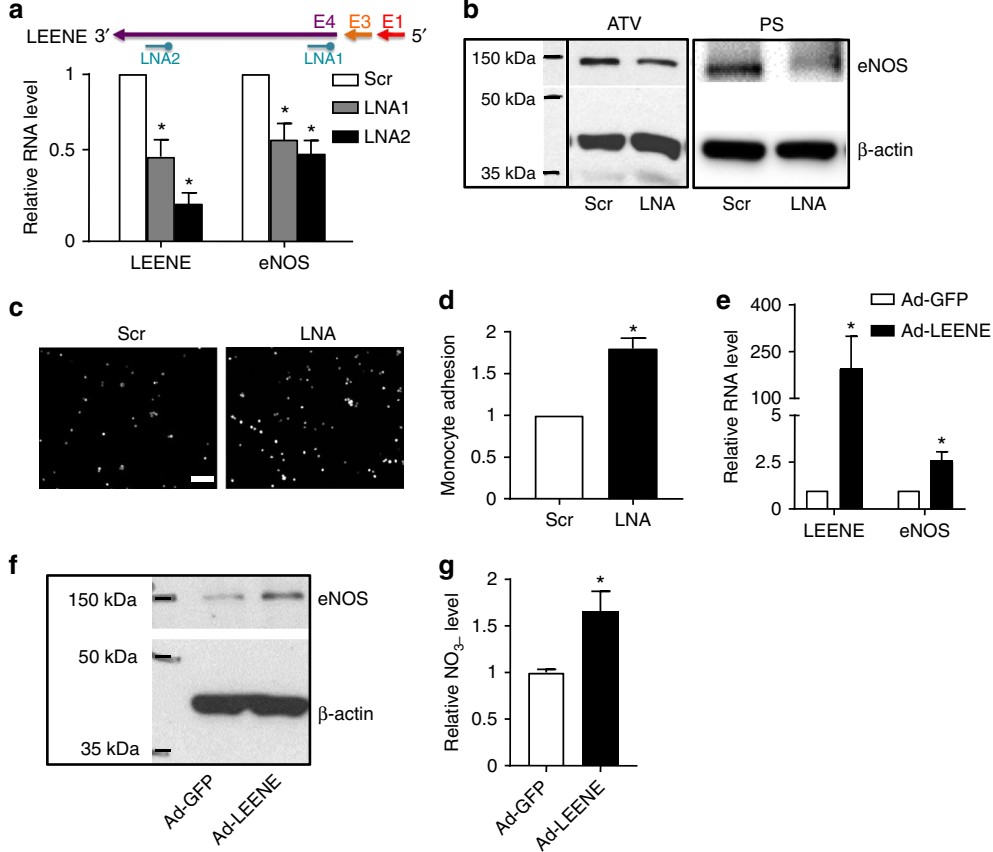

**Fig. 5** LEENE RNA regulates eNOS expression and EC function. **a–e** HUVECs were transfected with LNA (50 nM) targeting Exon 4 of LEENE. Basal RNA levels of LEENE and eNOS were detected by qPCR in **a**. Protein levels of eNOS in HUVECs treated with ATV or PS were revealed by immunoblotting. **c**, **d** ECs were transfected with scramble or LEENE LNA before subjected to PS for 12 h. Fluorescence-labeled THP-1 cells were added to the EC monolayer, and the monocytes adhering to ECs were visualized by fluorescence microscopy (scale bar = 100 μm). The representative images are shown in **c** and the quantification based on five randomly selected fields per group per experiment are shown in **d**. **e–g** HUVECs were infected with Ad-GFP or Ad-LEENE for 48 h. RNA levels of LEENE and eNOS were detected by qPCR (**e**), protein level of eNOS in HUVECs was revealed by immunoblotting (**f**), and NO production was measured by a fluorometric assay (**g**). Densitometry analysis of immunoblotting shown in **b** and **f** was performed (Supplementary Fig. 9). Data are presented as mean ± SEM. n = 3–5 in each group. Student's t test was used. *p < 0.05 compared to scrambled control or Ad-GFP in respective experiments

## Discussion

In this study, we took advantage of an integrative approach combining transcriptome and chromatin interactome profiling to identify LEENE, which is encoded by a distal enhancer region that forms proximity association with *eNOS* locus; its RNA transcripts enhance RNA Pol II binding to *eNOS* promoter and the consequent eNOS transcription. Inhibition of LEENE at either genomic (i.e., DNA) or transcriptional (i.e., RNA) level suppresses eNOS transcription, whereas overexpression of LEENE increases levels of eNOS and its derived NO bioavailability. Elucidation of this mechanism provides novel insights into the epigenetic modulation of endothelial gene expression in health and disease.

To identify the lncRNAs that potentially regulate eNOS transcription, we employed a systems biology approach to profile flow-regulated endothelial transcriptomes. Among all the lncRNAs that are differentially regulated by PS vs. OS, LEENE is ranked at the top (log$_2$FC = 1.93, correlation coefficient = 0.85) (Fig. 1b–d). Indeed, LEENE RNA was found to be regulated in concert with eNOS in ECs under hemodynamic, biochemical, and pharmacological stimuli (Fig. 1f, g). At the transcriptional level, LEENE and eNOS are co-regulated by KLF2 and KLF4 (Fig. 2). The hierarchical regulation of KLF2 and KLF4 upstream of LEENE and eNOS is reflected by the early induction of

KLF2/KLF4, preceding that of LEENE and eNOS (Fig. 1d). The identification of KLF2/KLF4-induced LEENE expands the repertoire of these TF-regulated transcriptional targets. In line with this notion, a recent report identified globally enriched TF-binding motifs for KLFs in ECs using ChIP-seq and ATAC-seq[40]. Therefore, KLFs may regulate a broader spectrum of transcriptional targets, including not only the protein-coding genes[28] and microRNAs[41, 42], but also lncRNAs.

While there have been extensive studies on the regulatory mechanisms of eNOS expression at multiple levels, there is a lack of information on the role of epigenetic modulation, particularly through lncRNAs and long-range DNA interaction. Summarizing the findings from our study, LEENE may enhance eNOS expression through two layers of regulation: (1) *LEENE* enhancer serves as a distal enhancer that forms proximity association with *eNOS* promoter (Figs. 3, 4); (2) in such chromosomal context, LEENE RNA transcripts induced by KLF2/KLF4 facilitate the binding of RNA Pol II to promote nascent RNA transcription of eNOS (Fig. 6). To tease out the reciprocal requirement of these two layers, we found that in ECs with *LEENE* enhancer ablated, overexpression of LEENE failed to induce eNOS expression (Supplementary Fig. 14a). Furthermore, in these *LEENE* enhancer-deleted cells, the association between LEENE RNA transcript and *eNOS* locus is significantly decreased under both

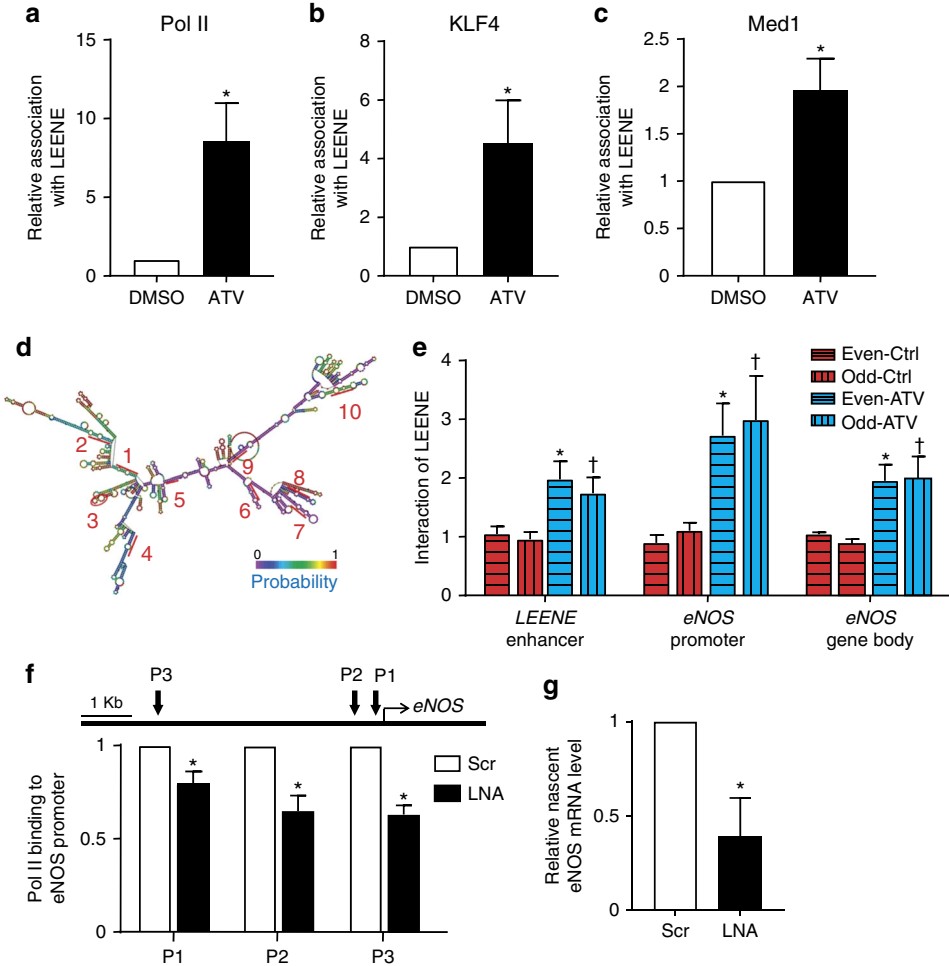

**Fig. 6** LEENE RNA promotes RNA Pol II binding and eNOS transcription. **a–c**, **e** HUVECs were treated with ATV for 24 h. The binding of RNA Pol II, KLF4, and MED1 to LEENE RNA was determined by RIP followed by qPCR (**a–c**). **d** Predicted secondary structure of LEENE RNA based on minimum free energy (MFE) and fragments complementary to ChIRP probes are labeled with numbers 1–10. Color scale shows the probabilities for every nucleotide to hold the structural position. Following ChIRP, interactions between LEENE RNA and respective DNA regions of *LEENE* and *eNOS* were detected by qPCR (in **e**). **f**, **g** Static ECs were transfected with scramble or LEENE LNA. The binding of RNA Pol II to *eNOS* promoter was determined by ChIP-qPCR with three primer sets flanking three regions upstream of *eNOS* TSS (in **f**). **g** Nascent RNA was captured in static ECs transfected with scramble or LEENE LNA. eNOS mRNA level was detected by qPCR. Data are presented as mean ± SEM, $n = 5$ in each group. *$p < 0.05$ compared with respective control in each experiment. In **a–c**, * denotes $p < 0.05$ compared with DMSO; in **e**, * indicates $p < 0.05$ between Ctrl and ATV using even-numbered probes; † denotes $p < 0.05$ between Ctrl and ATV using odd-numbered probes; in **f** and **g**, * means $p < 0.05$ between scramble vs. LNA groups. Student's $t$ test was applied

untreated and ATV-treated conditions (Supplementary Fig. 14b). These findings support the notion that the proximity association between *LEENE* and *eNOS* loci is a prerequisite for the association of LEENE RNA to *eNOS* promoter; without the *LEENE* enhancer region, LEENE RNA is not sufficient to enhance eNOS transcription. This hypothesis is illustrated in Fig. 8. It remains to be explored how the *LEENE* and *eNOS* loci come in proximity and form the promoter–enhancer contact in ECs, and whether LEENE RNA transcripts per se further stabilize such inter-chromosomal interaction.

Considering the genomic feature of *LEENE*, one may classify LEENE as an enhancer RNA (eRNA) as LEENE is encoded in a ~300 kb H3K27ac-enriched and H3K4me-enriched region. However, comparing to most eRNAs reported to promote nearby gene transcription in *cis*[43], LEENE does not seem to affect its neighboring genes, because (1) its RNA level is discordant with neighboring KTN1 and PELI2, which do not show differential expression in ECs subjected to different flow patterns or over-expression of KLF2/KLF4 (Supplementary Fig. 2a, b); and (2) neither LNA knockdown nor CRISPR deletion of *LEENE* locus

affects its adjacent KTN1 expression (Supplementary Fig. 2c, d). It is intriguing that LEENE transcription is highly concordant with eNOS, the key endothelial molecule encoded on chr 7. Despite the seemingly distinct chromosomal territories, *LEENE* and *eNOS* loci show proximity association in ECs, both under untreated or ATV/PS-treated conditions (Figs. 3e–g, 4c, d). This is in line with the emerging notion that lncRNAs may facilitate co-regulation of genes involved in similar biological processes[44]. Given the molecular mechanisms identified in this study, we reason that LEENE would be an example of lnc-eRNA or elncRNA, which is a transcript with initiation sites overlapping with enhancer regions and present in current lncRNA databases[19, 45].

Other than *eNOS*, *LEENE* may interact with genomic loci encoding a set of genes that are involved in multiple pathways crucial for endothelial homeostasis, e.g., cell adhesion and VEGF signaling (Supplementary Fig. 15). In addition, LEENE may also regulate other genes important for endothelial function through indirect mechanisms. For example, we found that LEENE LNA decreases, whereas LEENE overexpression increases

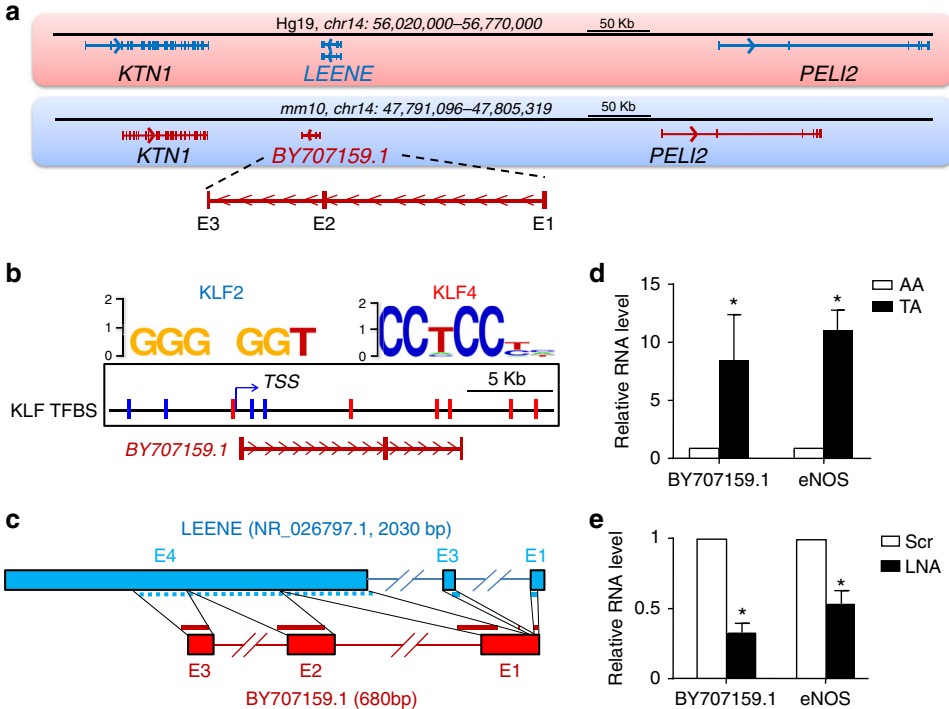

**Fig. 7** LEENE homolog in mouse. **a** Comparison of human *LEENE* and mouse *BY707159.1* loci. **b** Putative KLF2/4 TFBS in DNA region encoding *BY707159.1*, indicated by blue (for KLF2) and red (for KLF4). **c** Sequence alignment between human *LEENE* and *BY707159.1*. **d** Measurement of BY707159.1 RNA levels in thoracic aorta (TA) and aortic arch (AA) using qPCR. **e** qPCR of BY707159.1 and eNOS RNA level in isolated mouse lung ECs transfected with scramble or LEENE LNA. In each experiment, lungs from four animals were pooled for isolation and transfection. Data are average from four independent experiments. Error bars present mean ± SEM. *$p < 0.05$ between AA and TA in **d** and between scramble and LNA in **e** based on Student's *t* test

thrombomodulin (Tm), another KLF2 transcriptional target in ECs[46] (Supplementary Fig. 16). However in reference to the HUVEC Hi-C data, there is a lack of direct interaction between *LEENE* and *Tm* loci (Supplementary Fig. 17). The complete repertoire of LEENE-regulated transcriptome remains to be characterized.

In addition to LEENE, we also identified a number of other lncRNA loci in the eNOS 4C libraries, such as *MALAT1* and *MIAT* (Supplementary Fig. 18), all of which have been shown to be abundantly transcribed and play functional roles in ECs[15, 47]. The chromosomal contacts of other lncRNAs with eNOS may recruit additional chromatin remodelers to modulate eNOS transcription. These mechanisms may coordinate with TF binding and histone modifications to organize the chromatin conformation of eNOS, contributing to its transcriptional control. Given the recent study demonstrating the poor CpG content and the lack of flow-altered DNA methylation status in *eNOS* promoter[24], our findings suggest that the lncRNA-mediated chromatin remodeling may be an important factor other than DNA methylation in epigenetic regulation of eNOS, with both spatial and temporal control.

The cross-species conservation of lncRNAs is a challenging and key topic in the epigenetics field, as the estimated sequence homology between human and mouse lncRNAs is only 20%[48]. We identified BY707159.1, which is similar to human LEENE in several aspects including sequences, genomic structure, TFBS enrichment, differential regulation by flow patterns, and its gene regulation of eNOS (Fig. 7). Given the conservation of Firre in the repeating RNA domains[18], it is possible that LEENE is conserved between human and mouse in regions/domains important for its molecular function. Our findings set the stage for future systematic exploration of the functional roles of LEENE as an epigenetic player in cardiovascular health and disease.

## Methods

**Cell culture and shear stress experiments**. HUVECs (200p-05n, Cell Applications Inc., San Diego, CA) and HAoEC (304-05a, Cell Applications Inc., San Diego, CA) (Passages 6–8) from pooled donors were used in this study. The cells have been tested negative for mycoplasma contamination and prescreened to demonstrate stimulation-dependent angiogenesis and key EC signaling pathways. A parallel-plate flow system was used to impose shear stress to ECs cultured in flow channels by established methods as previously described[49]. This system is composed of a glass slide, a silicone gasket, and an acrylic plate chamber. A confluent monolayer of ECs was seeded onto the glass slide, and the flow channel space was created by sandwiching the gasket between the ECs and the chamber base. This assembly was then connected to a high reservoir, a low reservoir, and a peristaltic pump, thus mimicking circulation. The magnitude of applied shear stress ($\tau$) in such a flow system is governed by: $\tau = 6\eta Q/(h^2 w)$, where $\eta$ is the viscosity of perfusing media, $h$ and $w$ represent the height and width of the channel space, respectively, and $Q$ is the flow rate, which is determined by the height difference between the high and low reservoirs. Experimentally, a reciprocating syringe pump is connected to the circulating system to introduce a sinusoidal component with a frequency of 1 Hz (mimicking the pulse in the human body) onto the laminar shear stress, and the magnitude of oscillation can also be precisely controlled by adjusting the pump setup. The flow system was maintained at 37 °C and ventilated with 95% humidified air and 5% $CO_2$. Physiological flow with pulsatile shear stress (PS) and pathological flow with oscillatory shear stress (OS) were generated by circulating flow system and a reciprocating syringe pump and applied to ECs with a shear stress of $12 \pm 5$ and $0.5 \pm 5$ dyne cm$^{-2}$, respectively.

**Absolute quantification of lncRNA copy number**. The RNA copy number was determined following the protocol described by Tripathi et al[14]. In brief, in vitro transcribed LEENE RNA fragments were used as standards. LEENE RNA fragment ($10^9$ copies) and total RNA from $2 \times 10^4$ ECs were DNase I-treated, and subjected to reverse transcription. The reverse-transcribed LEENE complimentary DNA (cDNA) was serially diluted to generate a standard curve. The copy number of LEENE RNA in cDNA samples from $10^3$ cells were quantified in triplicates using Bio-Rad CFX Manager software.

**Transcription factor-binding sites analysis**. The KLF2 and KLF4 binding sites were predicted by using regular expression matches in R program based on the motifs from TRANSFAC database (Version 2015.4). Genomic regions were downloaded from the UCSC genome browser. KLF2 and KLF4 binding sites were

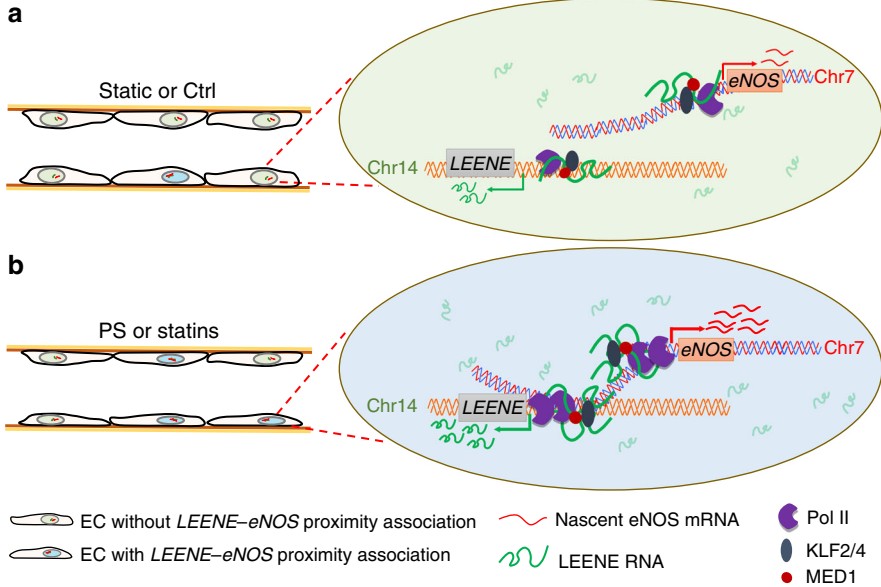

**Fig. 8** Schematic illustration of LEENE–eNOS regulatory mechanism. The *LEENE*-associated enhancer (located in chr14) forms proximity association with *eNOS* locus (in chr7) under both static/basal/control (ctrl) (**a**) and stimulated conditions (PS or statins) (**b**), but is at a higher probability in the latter condition. In both conditions, KLF2 and KLF4 transcriptionally regulate LEENE and eNOS through binding to TFBS in the promoters of both genes. The LEENE RNA transcripts serve as guides to facilitate RNA Pol II binding to the promoter of *eNOS*. This enhancer lncRNA-mediated transcriptional regulation positively modulates the nascent eNOS mRNA synthesis to promote endothelial function

predicted using DNA sequences from *LEENE* region spanning −20 to +5 kb of its TSS in hg19 and from *BY707159* region (chr14: 47,786,094–47,815,319, mm10).

**Analysis of Hi-C data**. Pre-processed 5-kb resolution inter-chromosomal hi-C matrices for HUVEC were accessed and downloaded online (GEO accession: GSE63525)[32, 33]. The *LEENE* genomic region was defined as chr14:56,240,000–56,290,000 in order to span the *LEENE* gene, the detected 4C signals, and the local H3K27ac marks. All genomic regions with non-zero signals in the inter-chromosomal matrices that were associated with the defined *LEENE* genomic region were located and annotated by gene (or closest gene if the region with detected signal was intergenic). These *LEENE*-associated genes were further filtered depending on whether they were detected to be differentially expressed in PS vs. OS at any time point.

**Library construction and analysis of 4C-seq**. The construction of 4C libraries was performed following previously published protocol[50]. Briefly, ECs were crosslinked with 2% formaldehyde, which was quenched with 0.1 M glycine. The cross-linked DNA underwent two rounds of digestion respectively by *Dpn*II and *Cvi*qI recognizing 4 bp restriction sites. Each digestion was followed by a reaction with T4 DNA ligase for proximity ligation. The resulting 4C template was used for the subsequent PCR reactions, of which 16 were pooled and purified for next-generation sequencing. The 4C sequencing reads were tested for the quality and aligned to human reference genome version hg19 by Bowtie 2[51]. Read numbers in given genomic location were counted by BEDTools[52] and normalized by the total mapped reads per sample. Circos plot were generated with RCircos[53] based on normalized read numbers. The 4C-seq raw data can be accessed from Gene Expression Omnibus with GEO accession number GSE103649.

**Subcellular fractionation and RNA isolation**. Subcellular fractionation was performed following published protocol[54] with minor modification. Briefly, HUVECs from three confluent 150 mm culture dishes were applied as independent triplicates. The cells were collected in 200 μl cold cytoplasmic lysis buffer (0.15% NP-40, 10 mM Tris pH 7.5, 150 mM NaCl) and incubated on ice for 5 min. The lysate was layered onto 500 μl cold sucrose buffer (10 mM Tris pH 7.5, 150 mM NaCl, 24% sucrose weight by volume) and centrifuged. The supernatant containing cytoplasmic component was quickly added to TRizol LS for RNA extraction. The nuclear pellet was gently suspended into 200 μl cold glycerol buffer (20 mM Tris pH 7.9, 75 mM NaCl, 0.5 mM EDTA, 50% glycerol, 0.85 mM DTT). An addition of cold nuclei lysis buffer (20 mM HEPES pH 7.6, 7.5 mM MgCl₂, 0.2 mM EDTA, 0.3 M NaCl, 1 M urea, 1% NP-40, 1 mM DTT) was added, followed by vortex and centrifuge. The supernatant containing nucleoplasmic fraction was mixed with TRizol LS for RNA extraction. Cold PBS (50 μl) was added to the remaining pellet and gently pipetted. After vigorous vortex to resuspend the chromatin, chromatin-associated RNA was extracted by adding 100 μl chloroform and TRizol reagent. RNA samples from three different fractions were dissolved with same amount of

RNase-free water and same volume of RNA was used for reverse-transcript and qPCR.

**CRIPSR-Cas9 gene editing**. We designed multiple sgRNA to target the genomic region of *LEENE* as illustrated in Fig. 4a. The sequences of sgRNAs are listed in Supplementary Table 3. The designed sgRNAs were sub-cloned into the CAS9-T2A-GFP-expression vector (Addgene: pX458) using designed *Bbs*I cloning site. All sgRNAs were tested with its cutting efficiency in HEK293 cells using the Surveyor Mutation Detection Kit from IDT (Supplementary Fig. 6).

**Cell transfection**. Two Antisense LNA GapmeRs specifically targeting two different regions of LEENE (NR_026797) were designed and purchased from Exiqon (Supplementary Table 4). siRNAs with scrambled or KLF2 targeting sequence were designed and purchased from Qiagen (SI03650318, SI04275110). LNAs or siRNA were separately transfected into ECs with Lipofectamine RNAiMAX following the protocol provided by the manufacturer. ECs were cultured for another 48 h after transfection before further analysis. Transfection of ECs with GFP-Cas9 with or without sgRNAs was performed with Cytofect HUVEC transfection kit (Cell Applications). Respective vectors (2 μg) were transfected per well of six-well plates, as the cells reached 80% confluency. After 1 h incubation with transfection mixture, antibiotics-free growth medium was added for another 48 h culture, before the cells were harvested.

**Monocyte adhesion assay**. Monocytes adhesion assay was performed as previously described[25]. THP-1 monocytes (ATCC) were labeled with CellTracker Green CMFDA Dye (Thermo Fisher #C2925) and incubated with monolayer ECs (4 × 10³ cells cm⁻²) for 15 min in a cell culture incubator. The non-attached THP-1 cells were then washed off with complete EC growth medium. The attached THP-1 cell numbers were evaluated on fluorescent microscopy using green fluorescent channel. Average numbers per sample were calculated from five randomly selected fields.

**Chromatin isolation by RNA purification**. ChIRP was performed following the protocols as described in the previous studies[55–57]. Biotin-labeled anti-sense oligo probes were designed and purchased from Biosearch Technologies (Supplementary Table 5) following several criteria: (1) number of probes = 1 probe per 100 bp of RNA length; (2) target GC = 45; (3) oligonucleotide length = 20 bp; (4) spacing length = 60–80 bp. The "even" and "odd" pools of probes were diluted into 100 μM concentration. After 24 h treatment with 1 μM ATV or DMSO, 1 × 10⁷ HUVECs were fixed with 1% glutaraldehyde for 10 min at room temperature. The pelleted cells were lysed and sonicated for 10 min using "30s ON, 30s OFF" program. The sonicated samples were then centrifuged and 1% of supernatant was taken as RNA input and DNA input, respectively. About 100 pmol probes were hybridized with supernatant at 37 °C for 4 h. Afterwards, washed Streptavidin-conjugated magnetic

beads were mixed with the reaction for another 30 min. Following several rounds of washing, beads were resuspended with 1 ml wash buffer and 100 µl mixture was taken for RNA isolation using TRIzol. The rest of the ChIRP precipitates underwent DNA isolation. qPCR analysis was performed to assess the RNA retrieval rate using β-actin as negative control and the LEENE-associated DNA sequences.

**RNA (RIP) and chromatin immunoprecipitation**. RIP was performed as previously described[58]. In general, after 24 h treatment with 1 µM Statin or DMSO, $1 \times 10^7$ HUVECs were washed with PBS, cross-linked by UV irritation (400 mJ cm$^{-2}$), and spun down by centrifuge. Whole cells were lysed with 500 µl lysis buffer (50 mM Tris, pH 7.5, 150 mM NaCl, 0.1% NP-40, 1 mM EDTA, and 100 units per ml RNAse inhibitor) and incubated overnight at 4 °C with 50 µl of Protein G dynabeads that were pre-washed and pre-mixed with antibodies or non-specific IgG control. Antibodies used for RIP assays include anti-RNA Pol II (mouse monoclonal to mouse and human RNA Pol II CTD repeat YSPTSPS, ab817, Abcam), anti-KLF4 (rabbit monoclonal to residues near the carboxy terminus of human KLF4 protein, 12173, Cell Signaling Technology), and anti-MED1 (rabbit polyclonal to residues between 1525 and carboxy terminus of MED1, A300-793A, Bethyl Laboratories). All of the antibodies have been previously authenticated for ChIP use[59–61] and we used 5 µl antibody for chromatins isolated from 1 mg input total protein. Following three times of wash to remove non-specific binding, RNA was extracted by TRizol and reverse transcribed for qPCR analysis.

ChIP assays were performed as previously described[62] using the same antibodies as RIP. Briefly, $1 \times 10^7$ HUVECs were treated with 0.75% formaldehyde for 20 min at room temperature. Afterwards, fixation was stopped by adding 125 mM glycine and cells were collected. The pelleted cells were lysed and sonicated for 4 min using "30s ON, 30s OFF" program at 4 °C. The sonicated samples were then centrifuged and 1% of supernatant was taken as input. After sonication, the chromatin was immunoprecipitated by various antibodies conjugated to pre-washed Protein A or Protein G Dynabeads. Protein and RNA were digested by proteinase K and RNase A, respectively. The purified chromatin DNA was then used as the template for a quantitative polymerase chain reaction. As an isotype control, non-specific IgG derived from the same species as specific antibodies were used in ChIP.

**DNA fluorescence in situ hybridization**. In-house probes detecting *eNOS* and *LEENE* genomic regions were generated from bacterial artificial chromosome (BAC) probes (Source BioScience LifeSciences). The clone IDs are *eNOS*, RP11-910F16 (length 183,744 bp) and *LEENE*, RP11-105H21 (length 183,093 bp). BAC probes were labeled by FISH tag DNA kit (Invitrogen). DNA FISH was performed following previously described protocols[63, 64]. Briefly, HUVECs were seeded on the coverslides and fixed directly with 4% formaldehyde and permeabilized with 0.1% saponin per 0.1% Triton X-100 in PBS for 10 min at room temperature. Cells were then equilibrated in 50% formamide per 2× SSC for 10 min at room temperature and denatured for 3 min at 78 °C. Afterwards, cells were hybridized overnight in a humidified chamber at 37 °C in 10 µl Hyb buffer (40% dextran sulfate plus 8×SSC) combined with 30 ng DNA FISH probes that have been freshly denatured at 78 °C for 5 min and cooled on ice. On the second day, the slides were washed three times with wash buffer (0.1% Tween plus 4× SSC). Cells were counter stained with DAPI, mounted with prolong buffer and imaged with Zeiss Apotome. The two probes were considered as proximally associated when the signals were completely overlapped or the distance between the centers of the signals was <1 µm, following the previous study[37]. Twenty pictures were randomly taken from each sample and three researchers were assigned to independently and blindly quantify the percentage of the cells showing proximity association. The mean values were used as the final result.

**NO bioavailability assay**. The NO production from HUVECs was detected as the accumulation of nitrate/nitrite by using a Nitrate/Nitrite Fluorometric Assay Kit (Cayman Chemical) as previously described[62]. Briefly, the phenol-red-free M199 medium used to culture ECs was collected and centrifuged. Fresh supernatant (at a volume of 20 µl) was used for NO assay. Nitrate was first converted to nitrite utilizing nitrate reductase, followed by DAN addition to form fluorescent product. The fluorescent signal was read using TECAN Infinite 200 pro (TECAN) under 360 nm excitation wavelength and 430 nm emission wavelength. The NO content was calculated based on the nitrate standard curve.

**Nascent RNA capture**. Newly synthesized mRNA species were isolated using Click-iT Nascent RNA Capture Kit (C10365, Invitrogen) according to manufacturer's protocol. Briefly, HUVECs were synchronized with 2% FBS in M199 medium for 8 h, followed by incubation in 0.2 mM of 5-ethymyluridine (EU, an alkyne-modified uridine analog, which is incorporated into the nascent RNA) for another 24 h, and total RNA was isolated using TRIzol reagent. A copper-catalyzed click reaction was performed using 5 µg RNA with 0.5 mM azide-modified biotin. The mixture was incubated at room temperature for 30 min following RNA precipitation. Biotin-labeled EU-RNA was then pulled down by mixing with Streptavidin T1 magnetic beads at room temperature for 30 min and the unbound RNA was washed away. The cDNA synthesis was performed directly on the beads using Superscript VILO cDNA synthesis kit (Invitrogen), followed by qPCR analysis.

**Quantitative PCR**. Reverse-transcription of RNA into cDNA was performed with PrimeScript RT Master Mix containing both Oligo dT primer and random 6mers primer (Takara Bio Inc.). KAPA SYBR FAST ROX Low supermix was used for qPCR following manufacturer's suggested protocol. All the primer sequences used were listed in Supplementary Tables 1 and 2.

**Western blot analysis**. Western blot was performed as previously described[3] using antibodies against eNOS (1:1000 dilution, rabbit polyclonal to the total human eNOS protein, 9572S, Cell Signaling Technology) and β-actin (rabbit monoclonal to mouse and human total β-actin protein, 8457S, Cell Signaling Technology, used at 1:4000 dilution) following standard protocol. Briefly, 20 µl of lysates from ECs was resolved on 8% SDS–PAGE, and proteins were transferred to PVDF membrane. Non-specific binding of antibody was blocked by washing with TBS buffer containing 10% milk for 1 h. After incubation with primary antibody overnight, the membrane was washed and incubated with HRP-conjugated secondary antibody for 1 h. The blots were visualized using the chemiluminescent detection method (Pierce). The levels of proteins present on the blots were quantified by densitometry using ImageJ (NIH). Uncropped scans of western blots are included in Supplementary Figs. 19, 20, and 21.

**LEENE homology analysis**. The sequence similarity between the predominant transcript of LEENE (NR_026797.1) and BY707159.1 was calculated by using EMBOSS Water tool, which was designed based on Smith–Waterman algorithm, with default parameters[65].

**Animal studies**. Animal study protocol was approved by Institutional Animal Care and Use Committee of City of Hope, Duarte. C57BL male mice (8-week-old) were randomly chosen and killed. TA and AA were isolated in PBS to tease out the perivascular adventitia. The vascular tissues were immediately snap-frozen in liquid nitrogen followed by RNA extraction with TRIzol reagent. Based on our previous experience, sample size was determined to have enough power to detect an estimated statistical difference between two groups. With a sample size of five in each group, this study can provide 80% powder to detect an effect size of 2 with 0.05 significant level using two-sided t-test between two groups. We did not expect large variation between two groups since the chosen animals are identical or similar regarding their age, gender, and background and raised under same condition. In the given case, no blinding was needed.

Mouse lung ECs isolation was performed following the protocol as previously described[62] with modifications. Specifically, monoclonal rat anti-mouse CD31 antibody from BD Transduction Laboratory (PECAM-1, clone MEC13.3) was covalently coupled to Dynabeads through overnight incubation (mix 1 µl of CD31 antibody with 20 µl of beads). For each experiment, the lungs from six C57BL male mice at the age of ~6 weeks were pooled. Lung tissue was excised, minced into 1 × 2 mm squares, digested with 2 mg per ml type I collagenase (Worthington) at 37 °C with gentle agitation for 45 min, triturated 12 times with a 30-cc syringe, and passed through a 70 µm disposable cell strainer (BD Falcon) into a 50 ml conical centrifuge tube. The cell suspension was then centrifuged at 400×g for 8 min at 4 °C, and the pellet was resuspended in 2 ml of medium. Anti-CD31-coated beads were added to the cell suspension, mixed, and incubated for 10 min at room temperature. The bead-bound cells were isolated with a magnet, resuspended in growth medium M199, and plated on collagen-coated T75 flasks. On the following day, the flasks were washed with medium to remove loosely adherent cells.

**Statistical analysis**. First, normal distribution from each group was confirmed using $\chi^2$ test before any comparison between groups. Statistical analysis was then performed using Student's t test (two-sided) between two groups or ANOVA followed by Bonferroni post test for multiple groups comparisons. If variances between two groups were significantly different (F-test), nonparametric Mann–Whitney U-test was applied. $p < 0.05$ was considered as statistically significant. As for all the experiments, at least three independent experiments were performed unless otherwise specified.

**Data availability**. All data supporting the current study are available in the article and its Supplementary Information Files or are available from the corresponding authors on request. The sequencing data sets have been deposited in the NCBI Gene Expression Omnibus (GEO) database (GEO accession number GSE103649).

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

## Acknowledgements

We would like to thank Drs. Arthur Riggs, Rama Natarajan, Amy Leung, Dustin Schones, Kevin Morris, and Jing-Kuan Yee at Beckman Research Institute, City of Hope, and Drs. John Y.-J. Shyy, Michael (Geoff) Rosenfeld, Bing Ren, and David Gorkin at UCSD for the useful discussions and their valuable suggestions, and Drs. Wouter de Laat and Adrien Melquiond for their consultation on 4C-seq and chromatin conformation

study. This work was supported in part by US NIH research grants K99/R00HL122368 (Z.C.), R01HL106579 and HL108735 (S.C. and S.S.), Beckman Research Institute Startup Fund (Z.C.), and American Heart Association Postdoctoral Fellowship 17POST33410101 (T.-S.H.). The Genome Editing and Analytical Cytometry Cores at City of Hope were supported by the National Cancer Institute of the National Institutes of Health under award number P30CA033572.

## Author contributions

Y.M., N.E.A., T.-S.H., C.-H.L., S.C., S.S., and Z.C. designed research; Y.M., N.E.A., T.-S.H., C.-H.L., S.L., J.K., H.M., Y.-T.W., and Z.C. performed research; Y.M., N.E.A., T.-S.H., F.-M.L., C.-H.L., M.R.M., S.G., S.S., and Z.C. analyzed the data; Y.M., N.E.A., T.-S.H., S.S., S.C., and Z.C. wrote and edited the manuscript. All authors have read and approved the manuscript.

## Additional information

**Competing interests:** The authors declare no competing financial interests.

