## [Peer Review File · Nature Communications]

Reviewers' Comments:

Reviewer #1:

Remarks to the Author:

In the present manuscript, authors have described the involvement of an lncRNA, LEENCR in positively influencing the expression of eNOS. Utilizing RNA seq data from endothelial cells subjected to physiological flow with pulsatile shear stress (PS) and pathological flow with oscillatory shear stress (OS) for different time points, authors identified several lncRNAs that are deregulated under OS and PS conditions. LEENCR lncRNA up regulated under PS conditions showed positive correlation with eNOS expression. High-C data reveals physical interaction between LEENCR enhancer and eNOS promoter. Further, authors demonstrate physical interaction between LEENCR RNA and KLF TF, Med1 and RNA pol II. LEENCR-depleted and LEENCR enhancer KO cells showed reduced levels of eNOS mRNA. Based on these data, authors conclude that LEENCR acts as a scaffold to enhance the physical association of LEENCR and eNOS genomic loci and further enhances the transcription of eNOS gene, potentially by recruiting TFs and mediator complexes to the promoter.

The ms lacks several important pieces of data that are required in order to support the model proposed by the authors.

Specific comments:

1. Overexpression of KLFs could result in the induction of LEENCR through indirect mechanisms. Authors should confirm the association of KLF2 & 4 on the promoter of LEENCR by ChIP-qPCR. In order to confirm that KLFs are the main contributor of LEENCR induction under PS conditions, authors should deplete KLFs, and test the induction of LEENCR upon PS. Alternatively, authors could make luciferase reporter constructs containing WT and KLF binding site-deleted mutant promoters of LEENCR, and see the effect on reporter expression under PS conditions.
2. What is the copy number of LEENCR under normal and PS conditions? Is this a poly A+ RNA? What is the coding potential of LEENCR?
3. For data presented in fig 3A & B, authors need to include a negative control, where the potential association between LEENCR to one of the genes located 100-200kb away from the LEENCR locus should be tested.
4. It is not clear whether the data presented in fig 4 (a-c; g and h) and supp. fig 6 utilized untreated or ATV-treated cells. Ideally, all these experiments should be under both the conditions.
5. In order to confirm that LEENCR RNA regulates the transcription of eNOS gene, authors need to perform RNA pol II ChIP (on eNOS promoter) and eNOS nascent transcript pull down assay in control versus LEENCR-depleted cells. To test whether LEENCR facilitates the recruitment of TF and Med1, they should also test the association of KLF and Med1 on the promoter of eNOS in LEENCR-depleted cells.
6. It is not clear why DNA-FISH in control cells does not show proximity association of both the alleles of LEENCR and eNOS? Does it mean that LEENCR regulates the expression of only one of the eNOS alleles?
7. In addition to data shown in fig 3G, authors should quantify the % of cells showing proximity association of one or both alleles of LEENCR and eNOS genes in control and KO cells. Also, DNA-FISH should be performed in control and ATV-treated EC cells to see if statin treatment increased the proximity association.
8. Finally, it is not obvious to me why authors have performed DNA-FISH to test the proximity association of LEENCR and eNOS genes in control and KO cells. Ideally, 3C followed by qPCR

should have done in control and KO cells to see the change in physical association between LEENCR coding region and eNOS promoter.

9. Promoter/enhancer KO cells showed reduced cytological proximity between LEENCR and eNOS3 genes, even though the transcript levels of both the genes showed only ~30% reduction. This could mean that the chromatin interaction between these two genes happens in a LEENCR transcript-independent manner. On the other hand, chromatin proximity facilitates LEENCR RNA to interact with the eNOS promoter, and possibly influences the recruitment of various TFs such as KLFs or mediators to eNOS promoter. Authors should perform ChIRP in control and enhancer KO cells (both under untreated and ATV-treated). Since KO cells do not show complete loss of LEENCR RNA (based on the data shown in Fig 4h), these experiments will tell whether proximity association between LEENCR and eNOS loci would be a prerequisite for the association of LEENCR RNA to eNOS promoter.

10. In order to determine whether the interaction between LEENCR-eNOS chromatin and/or LEENCR lncRNA association with eNOS promoter is responsible for the observed role of LEENCR, authors need to delete the coding region (instead of the enhancer) of LEENCR and determine the effect on LEENCR and eNOS chromatin association and also eNOS transcript levels in untreated and ATV-treated cells.

Authors could also try similar experiments in the LEENCR ASO-treated cells. In this case, if both the genes associate even in the absence of LEENCR RNA, then one could argue that LEENCR-eNOS gene interaction happens in an RNA-independent manner. However, such proximity association of genes would facilitate the recruitment of LEENCR to eNOS promoter/enhancer. Also, it needs to be determined whether cytological and molecular interactions between LEENCR and eNOS chromatin happen only under PS conditions.

11. To pinpoint the role of LEENCR as an RNA that recruits TF to eNOS promoter, authors could tether LEENCR to the promoter of eNOS, and test the effect on eNOS transcription and recruitment of TS such as KLF and mediators.

12. None of the data provided in ms actually supports authors' conclusion that LEENCR regulates the expression of eNOS through 'chromatin remodeling'. To me it looks like that LEENCR acts as a guide to facilitate the recruitment of one or more TFs or mediators to eNOS promoter. Obviously, authors need to perform TF ChIP in control and LEENCR KO cells to see effect on the recruitment of TS to eNOS promoter in presence or absence of LEENCR.

13. Authors have not provided any data supporting the role of LEENCR lncRNA in facilitating the chromatin association of LEENCR and eNOS genes. All of the data supports the involvement of LEENCR in regulating the steady state level of eNOS mRNA, probably via influencing eNOS transcription and/or mRNA stability. On the other hand, LEENCR enhancer KO data indicates that promoter/enhancer region of LEENCR and not the lncRNA itself could influence the chromatin interaction.

A good experiment to test authors claim that LEENCR RNA acts as scaffold to bring its own genomic locus to the promoter of eNOS (as mentioned in the discussion, page 15) would be to overexpress LEENCR in the promoter KO line or LEENCR KD cells, and show increased chromatin interaction between the two loci and enhanced eNOS transcription.

14. Finally, the model shown in Fig 7 is somewhat misleading. At present, there is no data indicating that the same LEENCR RNA interacts with both its own genomic locus as well as eNOS locus. Assays such as ChIRP, RAP always shows physical interaction between the nuclear-retained lncRNAs and their genomic loci. This could be due to the accumulation of newly transcribing lncRNA transcripts at their genomic loci. The model should be modified to show the existence of separate LEENCR/TF/mediator complexes on LEENCR and eNOS chromatin. In addition, they could show the physical association of these two genomic loci.

Reviewer #2:

Remarks to the Author:

This paper identifies the lncRNA, LEENCR, as a regulator of eNOS gene expression. The authors use multiple genomic approaches to suggest that physiological flow induces KLF2/KLF4 and these TF activate the expression of LEENCR, which serves as a scaffold to enhance KLF2 dependent eNOS transcription. These data are innovative and experiments well conducted. My specific concerns are as follows:

1. Since KLF2 induces LEENCR, it is important to determine if other KLF2 dependent genes are regulated by LEENCR such as thrombomodulin or other flow regulated TF such as SREBP 2. In addition, it is likely that other interactions must take place since KLFs and LEENCR are expressed in non- endothelial cells.
2. Can expression of eNOS promoter confer LEENCR specific regulation of eNOS in cells that typically lack eNOS (HEK cells or fibroblasts) ?
3. How does a reduction in LEENCR in Fig 4a reduce basal eNOS mRNA levels by 50%? Does a reduction in LEENCR attenuate KLF2 mediated eNOS expression induced by flow ? Does the loss of LEENCR affect other KLF2 dependent genes in EC and non-EC ?
4. The levels of eNOS protein should be shown in Fig 5, which would reflect more mRNA.
5. The authors are fortunate that this LNC is conserved in mice. Some data show that LEENCR gapmers reduce eNOS mRNA levels in mouse EC would be critical to show similarity of functions for this LNC RNA.

Referee 1:

We appreciate the insightful comments and suggestions from the Referee and we have performed new experiments to provide several important pieces of data. Accordingly, we have revised the results, discussion, and conclusion and modified the model previously proposed. We believe the manuscript has been significantly improved. Of note, because of the new mechanism we have identified during the revision, we have changed the name LEENCNCR (lncRNA that enhances eNOS through chromatin remodeling) to LEENE (lncRNA that enhances eNOS expression) and we have uniformly used LEENE as the acronym throughout the point-to-point response letter and the revised manuscript. All changes in the manuscript are marked in red. The detailed point-to-point responses are outlined below:

1. Overexpression of KLFs could result in the induction of LEENCNCR through indirect mechanisms. Authors should confirm the association of KLF2 & 4 on the promoter of LEENCNCR by ChIP-qPCR. In order to confirm that KLFs are the main contributor of LEENCNCR induction under PS conditions, authors should deplete KLFs, and test the induction of LEENCNCR upon PS. Alternatively, authors could make luciferase reporter constructs containing WT and KLF binding site-deleted mutant promoters of LEENCNCR, and see the effect on reporter expression under PS conditions.

In accordance to the referee's comments, we have performed KLF4 ChIP-qPCR to confirm the association of KLF4 on the promoter of LEENE. Anti-KLF4 antibody (ChIP grade) was used to precipitate KLF4 and the associated chromatin DNA from HUVECs infected with Ad-Null or Ad-KLF4 viruses. There was a robust binding between KLF4 and multiple regions in ~4 kb upstream of LEENE TSS; this interaction was significantly increased by Ad-KLF4 (see figure below). We did not perform the KLF2 ChIP because there is currently no good antibody against KLF2 with ChIP grade.

To confirm the role of KLF as a main contributor of LEENE induction under PS condition, we have knocked down KLF2 using siRNA in ECs subjected to PS, as KLF2 is regarded as the most important signal-dependent TF in ECs. As shown in the figure below, KLF2 knockdown significantly reduced LEENE level in the PS-imposed ECs.

We have included these data in the revised Figs. 1 and 2.

The luciferase reporter assay would also be a nice experiment, and we will try to perform it in future studies.

2. *What is the copy number of LEENCR under normal and PS conditions? Is this a poly A+ RNA? What is the coding potential of LEENCR?*

We have performed absolute quantification with qPCR to determine the LEENE copy number in HUVECs. Based on three batches of cells, LEENE is transcribed at ~10 copies/cell in ECs under static condition and hence ~40 copies/cell under PS condition. It is a poly A⁺ RNA, which was identified from RNA-seq profiling obtained with poly A-selected RNA libraries. Based on FANTOM5, LEENE does not have any coding potential. We have included this information in the revised manuscript (Line 23, Page 6 to Line 2, Page 7).

3. *For data presented in fig 3A & B, authors need to include a negative control, where the potential association between LEENCR to one of the genes located 100-200kb away from the LEENCR locus should be tested.*

We have included a negative control as the referee suggested. Specifically, we tested the association between LEENE RNA and two regions located ~150 kb up- (*PELI2*) and downstream (*KTN1*) of the *LEENE* locus, respectively. As shown below, neither region showed significant increase in LEENE interaction under statin treatment. We have included this data in the revised Supplemental Fig. 12d.

4. *It is not clear whether the data presented in fig 4 (a-c; g and h) and supp. fig 6 utilized untreated or ATV-treated cells. Ideally, all these experiments should be under both the conditions.*

Pursuant to the referee's suggestion, we have clarified the conditions of data presented in the original Fig. 4 and Suppl. Fig. 6. Furthermore, we have repeated and added several key experiments, including ChIRP-qPCR detection of LEENE-associated eNOS DNA, and FISH analysis for proximity of eNOS and LEENE loci in ECs with LEENE-KO (enhancer or coding region, respectively) or LNA under both untreated and statin-treated conditions. These newly acquired data are included in the revised Figs. 3-6.

5. *In order to confirm that LEENCR RNA regulates the transcription of eNOS gene, authors need to perform RNA pol II ChIP (on eNOS promoter) and eNOS nascent transcript pull down assay*

in control versus LEENCR-depleted cells. To test whether LEENCR facilitates the recruitment of TF and Med1, they should also test the association of KLF and Med1 on the promoter of eNOS in LEENCR-depleted cells.

We have performed the experiments according to the excellent suggestion from the referee. As shown in the figure below, compared with ECs transfected with scramble control, LEENE LNA resulted in a significant decrease in nascent eNOS transcript (revealed by nascent RNA pulldown combined with qPCR in Panel A) as well as the association between RNA Pol II and eNOS promoter region (determined by ChIP-qPCR with multiple primer sets detecting three fragments in the eNOS promoter region, Panel B). However, based on KLF4 and Med1 ChIP-qPCR data, LEENE LNA did not cause significant difference in the association between these proteins and eNOS promoter regions (Panels C and D). These data suggest that LEENE RNA regulates the transcription of eNOS gene by facilitating the function of RNA Pol II and nascent RNA transcription, without altering the recruitment of KLF and Med1. We have included these data in the revised Fig. 6 and Supplemental Fig. 13.

6. It is not clear why DNA-FISH in control cells does not show proximity association of both the alleles of LEENCR and eNOS? Does it mean that LEENCR regulates the expression of only one of the eNOS alleles?

Indeed, we do not see the proximity association of both alleles of LEENE and eNOS in all the cells. In most of the cells (~500 cells per biological replicate, 2-3 replicates per experimental group) that we have imaged with DNA-FISH, we observed proximity association of only one of the two alleles. In a small portion (<1%) of cells, we did observe proximity association of both alleles of LEENE and eNOS. This probably means that there is a much lower chance that *LEENE* interacts with both eNOS alleles at the same time point in ECs harvested for DNA-FISH. However, this does not exclude the possibility that both *LEENE* alleles regulate the expression of two *eNOS* alleles.

7. In addition to data shown in fig 3G, authors should quantify the % of cells showing proximity association of one or both alleles of *LEENCR* and *eNOS* genes in control and KO cells. Also, DNA-FISH should be performed in control and ATV-treated EC cells to see if statin treatment increased the proximity association.

We have performed DNA-FISH in control and ATV-treated EC cells and quantified the percentage of cells showing proximity association (distance < 1 μm) of one or both alleles of *LEENE* and *eNOS* genes. As shown below, in ECs transfected with control Cas9 vector, statin increased the percentage of cells with *LEENE*-*eNOS* proximity association. Under both untreated and treated conditions, *LEENE* KO (Cas9+sgRNA) decreased the percentage of cells with this association. We have included these new data in the revised Fig. 4.

8. Finally, it is not obvious to me why authors have performed DNA-FISH to test the proximity association of *LEENCR* and *eNOS* genes in control and KO cells. Ideally, 3C followed by qPCR should have been done in control and KO cells to see the change in physical association between *LEENCR* coding region and *eNOS* promoter.

Because DNA-FISH has been commonly used to validate the chromatin interaction revealed by 3C based methods (Giorgetti and Heard, *Genome Biol.* 2016), we performed DNA-FISH to validate the proximity association of *LEENE* and *eNOS* genes, as well as its regulation by *LEENE* enhancer. As shown in the figure above, *LEENE* enhancer KO cells have decreased association between *LEENE* and *eNOS*.

In line with the referee's suggestion, we have tried to perform 3C followed by qPCR, but we were unable to detect the *eNOS* and *LEENE* DNA interaction using this method. In consultation with Dr. Wouter de Laat (Hubrecht Institute, the Netherlands), the senior author of Hagege et al. Quantitative analysis of chromosome conformation capture assays (3C-qPCR), *Nat. Protocol*, 2007, we were advised that "3C technology is particularly suited to identify chromatin loops formed in genomic regions of up to several hundreds of kilobases in size, but less suitable for inter- or trans-chromosomal interaction", which is the case in our study. We hope the referee would concur with this technical problem.

9. Promoter/enhancer KO cells showed reduced cytological proximity between *LEENCR* and

eNOS3 genes, even though the transcript levels of both the genes showed only ~30% reduction. This could mean that the chromatin interaction between these two genes happens in a *LEENCR* transcript-independent manner. On the other hand, chromatin proximity facilitates *LEENCR* RNA to interact with the *eNOS* promoter, and possibly influences the recruitment of various TFs such as KLFs or mediators to *eNOS* promoter. Authors should perform ChIRP in control and enhancer KO cells (both under untreated and ATV-treated). Since KO cells do not show complete loss of *LEENCR* RNA (based on the data shown in Fig 4h), these experiments will tell whether proximity association between *LEENCR* and *eNOS* loci would be a prerequisite for the association of *LEENCR* RNA to *eNOS* promoter.

We appreciate the referee's insightful comments. Accordingly, we have performed ChIRP in control and enhancer KO cells (both untreated and ATV-treated). As shown in the figure below, the association between *LEENE* transcript and *eNOS* locus was significantly decreased in *LEENE* enhancer KO cells, under both conditions. This finding supports the notion that "proximity association between *LEENE* and *eNOS* loci would be a prerequisite for the association of *LEENE* RNA to *eNOS* promoter". We have included these data in revised supplemental Fig. 14 and revised the discussion in the manuscript accordingly (Lines 10-12, Page 16).

10. In order to determine whether the interaction between *LEENCR*-*eNOS* chromatin and/or *LEENCR* lncRNA association with *eNOS* promoter is responsible for the observed role of *LEENCR*, authors need to delete the coding region (instead of the enhancer) of *LEENCR* and determine the effect on *LEENCR* and *eNOS* chromatin association and also *eNOS* transcript levels in untreated and ATV-treated cells. Authors could also try similar experiments in the *LEENCR* ASO-treated cells. In this case, if both the genes associate even in the absence of *LEENCR* RNA, then one could argue that *LEENCR*-*eNOS* gene interaction happens in an RNA-independent manner. However, such proximity association of genes would facilitate the recruitment of *LEENCR* to *eNOS* promoter/enhancer. Also, it needs to be determined whether cytological and molecular interactions between *LEENCR* and *eNOS* chromatin happen only under PS conditions.

Following these excellent suggestions, we have examined the interaction between *LEENE* and *eNOS* chromatin in *LEENE* LNA/ASO-transfected ECs using DNA FISH. As shown in Panels A and B in the figure below, *LEENE* LNA did not result in significant change in *LEENE*-*eNOS*

proximity association, in both statin-treated and untreated conditions. These findings suggest that LEENE-eNOS gene interaction can indeed happen in an RNA-independent manner.

We have also constructed a new CRISPR-cas9 vector with sgRNAs targeting specifically the LEENE coding region. With this vector, we have investigated the effect of coding region deletion on eNOS transcript levels. In Panel C below, qPCR analysis demonstrated that ablation of the LEENE coding region, which in turn inhibited LEENE transcription, decreased eNOS mRNA level in statin-treated ECs. Together with our data obtained with LEENE LNA, these findings suggest that LEENE RNA is important for eNOS expression; however, the LEENE RNA is not required for LEENE-eNOS genomic interaction.

We have also determined whether cytological and molecular interactions between LEENE and eNOS chromatin happen **only** under PS conditions. We have performed 4C with eNOS as the viewpoint and also DNA-FISH in ECs under static or PS conditions. Both experiments revealed that the LEENE-eNOS interaction could also happen in static/untreated conditions.

11. To pinpoint the role of *LEENCR* as an RNA that recruits TF to eNOS promoter, authors could tether *LEENCR* to the promoter of eNOS, and test the effect on eNOS transcription and recruitment of TS such as KLF and mediators.

We appreciate the referee's suggestion. To tether LEENE to the promoter of eNOS, we will need to use sgRNA-guided Cas9 to cleave eNOS promoter and introduce LEENE sequence (~2 kb) using a donor template. Alternatively, we could utilize the BoxB tethering system as described in Wang et al Nature 2011 and Li et al Nature 2014. We will try to perform such experiment in our future studies. To partially address the role of LEENE as an RNA to recruit TF to eNOS promoter, we have overexpressed LEENE in HUVECs and performed eNOS nascent transcript pulldown (transcription).

As shown in the right figure, overexpression of LEENE increased the transcription of nascent eNOS mRNA, which supports that LEENE as an RNA facilitate eNOS transcription in ECs.

12. None of the data provided in ms actually supports authors' conclusion that LEENCR regulates the expression of eNOS through 'chromatin remodeling'. To me it looks like that LEENCR acts as a guide to facilitate the recruitment of one or more TFs or mediators to eNOS promoter. Obviously, authors need to perform TF ChIP in control and LEENCR KO cells to see effect on the recruitment of TF to eNOS promoter in presence or absence of LEENCR.

The referee's critiques are well taken. As provided in the response to the referee's question #5, LEENE-depleted ECs showed a decreased binding in RNA Pol II, but not that of KLF4 or MED1 to eNOS promoter. Therefore, in line with the referee's comments, LEENE likely acts as a guide to facilitate the recruitment of RNA Pol II to the eNOS promoter, but not through chromatin remodeling. We have therefore changed the name to LEENCR to LEENE. Based on these new results, we have revised our Results (Lines 8-16, Page 13) and Discussion (Lines 6-7, Page 16).

13. Authors have not provided any data supporting the role of LEENCR lncRNA in facilitating the chromatin association of LEENCR and eNOS genes. All of the data supports the involvement of LEENCR in regulating the steady state level of eNOS mRNA, probably via influencing eNOS transcription and/or mRNA stability. On the other hand, LEENCR enhancer KO data indicates that promoter/enhancer region of LEENCR and not the lncRNA itself could influence the chromatin interaction.

A good experiment to test authors claim that LEENCR RNA acts as scaffold to bring its own genomic locus to the promoter of eNOS (as mentioned in the discussion, page 15) would be to overexpress LEENCR in the promoter KO line or LEENCR KD cells, and show increased chromatin interaction between the two loci and enhanced eNOS transcription.

We have taken the excellent suggestion from the referee and performed qPCR in HUVECs first transfected with a CRISPR vector targeting LEENE enhancer and then infected with Ad-GFP and Ad-LEENE. As shown in the figure below (Panel A), in cells with LEENE enhancer region ablated, overexpression of LEENE failed to induce eNOS expression. This suggests that, without the LEENE enhancer region, LEENE RNA is not sufficient to enhance eNOS transcription. Therefore, we have removed the discussion in the earlier version of manuscript that "LEENE RNA acts as scaffold to bring its own genomic locus to the promoter of eNOS" and we have revised the Discussion (Lines 3-18, Page. 16) and included these data in Supplemental Fig. 14b. In line with the referee's comment, we have now provided data supporting "the involvement of LEENE in regulating the level of eNOS mRNA, via influencing eNOS transcription".

We have also tested the possibility that LEENE regulates the mRNA stability of eNOS by treating LNA-transfected ECs with actinomycin D, a transcription inhibitor. As shown in the figure below (Panel B), LEENE LNA did not significantly inhibited the eNOS mRNA stability, indicating that the post-transcriptional regulation is unlikely to be the mechanism underlying LEENE regulation of eNOS.

14. Finally, the model shown in Fig 7 is somewhat misleading. At present, there is no data indicating that the same LEENCR RNA interacts with both its own genomic locus as well as eNOS locus. Assays such as ChIRP, RAP always shows physical interaction between the nuclear-retained lncRNAs and their genomic loci. This could be due to the accumulation of newly transcribing lncRNA transcripts at their genomic loci. The model should be modified to show the existence of separate LEENCR/TF/mediator complexes on LEENCR and eNOS chromatin. In addition, they could show the physical association of these two genomic loci.

We have modified the schematic model according to this critique and the newly obtained data in the revised Fig. 8.

Referee 2:

This paper identifies the lncRNA, LEENCR, as a regulator of eNOS gene expression. The authors use multiple genomic approaches to suggest that physiological flow induces KLF2/KLF4 and these TF activate the expression of LEENCR, which serves as a scaffold to enhance KLF2 dependent eNOS transcription. These data are innovative and experiments well conducted.

We are extremely appreciative of Referee's valuable feedback and have addressed the specific concerns. Of note, because of the new mechanism we have identified during the revision, we have changed the name of LEENCR (lncRNA that enhances eNOS through chromatin remodeling) to LEENE (lncRNA that enhances eNOS expression) and we have uniformly used LEENE as the acronym throughout this point-to-point response letter and the revised manuscript. All changes in the manuscript are marked in red.

1. Since KLF2 induces LEENCR, it is important to determine if other KLF2 dependent genes are regulated by LEENCR such as thrombomodulin or other flow regulated TF such as SREBP2. In addition, it is likely that other interactions must take place since KLFs and LEENCR are expressed in non-endothelial cells.

We have detected thrombomodulin (Tm) and SREBP2 in ECs transfected with LEENE LNA or infected with Ad-LEENE. Compared with the respective controls, Tm mRNA levels were decreased by LEENE LNA, while increased by Ad-LEENE (Panels A and B below). Although SREBP2 showed a slight trend of increase in response to LEENE LNA, this was not statistically significant (Panel C). We have included these data in the Supplemental Fig. 16 and discussed on this point in Lines 14-19, Page 17.

To evaluate whether other interactions take place in non-EC cells, we checked Hi-C data collected from human epithelial cells and MCF7 cells (Barutcu et al. Genome Biol. 2015) for chromosomal interaction between LEENE and other KLF target genes, such as Tm, NFE2L2 (Nrf2), HMOX1, and NQO1. Unlike ECs, we did not observe any signals revealing proximity of LEENE with regions encoding these genes. Therefore, in line with the referee's comment, it is likely that other interactions do take place since KLFs and LEENCR are expressed in non-endothelial cells.

2. Can expression of eNOS promoter confer LEENCR specific regulation of eNOS in cells that typically lack eNOS (HEK cells or fibroblasts)?

To address this question, we first performed qPCR to determine the relative levels LEENE, KLF2, KLF4, and eNOS in HEK293 cells in comparison to HUVECs. As demonstrated in the table below, HEK293 cells express lower levels of LEENE, KLF2, and eNOS (indicated by higher Cq values).

	HUVECs (Cq value)	HEK293 (Cq value)
LEENE	29	33
KLF2	25	31
KLF4	29	29
eNOS	24	33

When we overexpressed LEENE in HEK293 cells, eNOS mRNA was still detected only at a negligible level. We also overexpressed LEENE in HEK293 cells transfected with luciferase reporter driven by eNOS promoter. As shown in the figure to the right, we only detected a slight but insignificant induction of luciferase activity. Therefore, the (over)expression of eNOS promoter per se cannot confer LEENE specific regulation of eNOS in non-eNOS expressing cells.

3. How does a reduction in LEENCR in Fig 4a reduce basal eNOS mRNA levels by 50%? Does a reduction in LEENCR attenuate KLF2 mediated eNOS expression induced by flow? Does the loss of LEENCR affect other KLF2 dependent genes in EC and non-EC?

To explore the mechanism underlying the LEENE regulation of eNOS at the basal level, we performed eNOS nascent RNA pulldown assay and ChIP analysis to detect the association between RNA Pol II and eNOS promoter region in ECs with LEENE knockdown using LNA. As shown in Panels A and B, LEENE LNA decreased significantly the nascent eNOS mRNA level as well as the association between RNA Pol II and eNOS promoter (P1-P3 represent 3 different regions from the eNOS promoter). These data suggest that LEENE acts as a guide to facilitate the recruitment of RNA Pol II to the eNOS promoter. We have included these newly acquired data in Fig. 6 and discussed this mechanism in Discussion (Lines 3-7, Page 16).

We have detected eNOS expression under PS conditions in ECs transfected with scramble or LEENE LNA. As shown in the figure below, knockdown of LEENE using either LNAs significantly decreased eNOS expression in PS-imposed ECs. These data are included in the revised Fig. 5.

As indicated in the response to question #1, knockdown of LEENE decreases Tm in ECs. However, as shown in the bar graph below, LEENE LNA did not affect Nrf2, another KLF2-regulated gene in ECs. We have also tested the effect of LEENE knockdown in MCF-7, a non-EC line. Inhibition of LEENE did not significant affect other KLF2-dependent genes in MCF7. We have included these new data in the Supplemental Fig. 16.

4. The levels of eNOS protein should be shown in Fig 5, which would reflect more mRNA.

We have performed immunoblotting for eNOS protein levels which indeed reflect more mRNA in Ad-LEENE-infected ECs. We have updated Fig. 5 accordingly.

5. The authors are fortunate that this LNC is conserved in mice. Some data show that LEENCR gapmers reduce eNOS mRNA levels in mouse EC would be critical to show similarity of functions for this LNC RNA.

We have performed experiment in which we transfected LEENE LNA gapmers to mouse lung ECs. As shown in figure to the right, LEENE LNA decreased eNOS mRNA levels in mouse EC (lung ECs from 5-6 mice/group, three independent experiments). We have included these data in the revised Fig. 7.

Reviewers' Comments:

Reviewer #1:

Remarks to the Author:

The authors have addressed most of my concerns by including newer data. I now recommend this ms for publication in NC.

Reviewer #2:

Remarks to the Author:

All of my comments have been addressed.